# Coral high molecular weight carbohydrates support opportunistic microbes in bacterioplankton from an algae-dominated reef

Bianca M. Thobor,[1] Andreas F. Haas,[2] Christian Wild,[1] Craig E. Nelson,[3] Linda Wegley Kelly,[4] Jan-Hendrik Hehemann,[5,6] Milou G. I. Arts,[2] Meine Boer,[2] Hagen Buck-Wiese,[5,6] Nguyen P. Nguyen,[5,6] Inga Hellige,[5,6] Benjamin Mueller[1,3,7,8]

**ABSTRACT** High molecular weight (HMW; >1 kDa) carbohydrates are a major component of dissolved organic matter (DOM) released by benthic primary producers. Despite shifts from coral to algae dominance on many reefs, little is known about the effects of exuded carbohydrates on bacterioplankton communities in reef waters. We compared the monosaccharide composition of HMW carbohydrates exuded by hard corals and brown macroalgae and investigated the response of the bacterioplankton community of an algae-dominated Caribbean reef to the respective HMW fractions. HMW coral exudates were compositionally distinct from the ambient, algae-dominated reef waters and similar to coral mucus (high in arabinose). They further selected for opportunistic bacterioplankton taxa commonly associated with coral stress (i.e., *Rhodobacteraceae*, *Phycisphaeraceae*, *Vibrionaceae*, and *Flavobacteriales*) and significantly increased the predicted energy-, amino acid-, and carbohydrate-metabolism by 28%, 44%, and 111%, respectively. In contrast, HMW carbohydrates exuded by algae were similar to those in algae tissue extracts and reef water (high in fucose) and did not significantly alter the composition and predicted metabolism of the bacterioplankton community. These results confirm earlier findings of coral exudates supporting efficient trophic transfer, while algae exudates may have stimulated microbial respiration instead of biomass production, thereby supporting the microbialization of reefs. In contrast to previous studies, HMW coral and not algal exudates selected for opportunistic microbes, suggesting that a shift in the prevalent DOM composition and not the exudate type (i.e., coral vs algae) *per se*, may induce the rise of opportunistic microbial taxa.

**IMPORTANCE** Dissolved organic matter (DOM) released by benthic primary producers fuels coral reef food webs. Anthropogenic stressors cause shifts from coral to algae dominance on many reefs, and resulting alterations in the DOM pool can promote opportunistic microbes and potential coral pathogens in reef water. To better understand these DOM-induced effects on bacterioplankton communities, we compared the carbohydrate composition of coral- and macroalgae-DOM and analyzed the response of bacterioplankton from an algae-dominated reef to these DOM types. In line with the proposed microbialization of reefs, coral-DOM was efficiently utilized, promoting energy transfer to higher trophic levels, whereas macroalgae-DOM likely stimulated microbial respiration over biomass production. Contrary to earlier findings, coral- and not algal-DOM selected for opportunistic microbial taxa, indicating that a change in the prevalent DOM composition, and not DOM type, may promote the rise of opportunistic microbes. Presented results may also apply to other coastal marine ecosystems undergoing benthic community shifts.

Address correspondence to Bianca M. Thobor, thobor@uni-bremen.de.

The authors declare no conflict of interest.

See the funding table on p. 21.

**KEYWORDS** coral-algae phase shift, carbohydrates, 16S rRNA sequencing, microbial metabolism, microbialization, opportunism

Corals are the main ecosystem engineers of tropical coral reefs, as they provide habitat and nutrients (1) to one of the most diverse and productive ecosystems on the planet (2, 3). However, coral cover is declining on many reefs worldwide due to global and local human stressors (4), often leading to overgrowth by fleshy algae (5–8). Particularly in the Caribbean widespread shifts toward stages of fleshy macroalgae dominance have been reported (7, 9, 10). These shifts have been associated with changes in coral reef community metabolism caused by benthic primary producer-specific differences in the exudation of dissolved organic matter (DOM) (11).

Algae usually exude more DOM than corals which increases the bacterioplankton abundance in reef water (12, 13). In addition, algae DOM stimulates microbial respiration (11) which can lead to deoxygenation of reefs (14–16). Concomitantly, less energy is transformed into microbial biomass (i.e., a shift from biomass generation to respiration), reducing the transfer of energy to higher trophic levels (17, 18). This shift in ecosystem-wide energy allocation from heterotrophic macrobes (e.g., fish and invertebrates) to foremost microbes was termed the microbialization of reefs and was proposed to occur globally on degraded, algae-dominated reefs (17, 19). Algae DOM also appears to select for putative opportunistic and pathogenic microbes (17, 20–22). Combined, these indirect effects of algae DOM on the microbial community can lead to coral mortality through hypoxia and disease (23–25). Coral mortality opens up space on the reef for algae growth, thus resulting in the DDAM positive feedback loop (DOC, disease, algae, and microorganisms) which is considered to facilitate reef degradation (26, 27).

These contrasting responses of microbial communities to coral- vs algae DOM suggest underlying differences in DOM composition. Indeed, liquid chromatography-tandem mass spectrometry has recently revealed a great compositional diversity in coral and algae exudates (28, 29). However, this method is mostly limited to low molecular weight (LMW) components (e.g., organic acids, lipids, and lipid-like molecules) which efficiently elute from solid phase extraction columns (30, 31), thus not capturing most carbohydrates. Carbohydrates are the most abundant biomolecules of high molecular weight (HMW; i.e., >1 nm or 1,000 Dalton) DOM in surface oceans (32, 33) and play a major role in shaping bacterioplankton communities (34, 35). Only very few studies have investigated the carbohydrate composition of the coral reef DOM so far. Nelson et al. (22) found increased concentrations of fucose, galactose, and rhamnose in bulk macroalgae-DOM (i.e., HMW + LMW), which exerted strong effects on the composition of natural bacterioplankton communities (22). Particularly the growth of copiotrophs and putative pathogens belonging to *Gammaproteobacteria* was stimulated. In contrast, bulk DOM exuded by the coral *Porites lobata* mainly consisted of glucose, mannose, and xylose and was similar in composition to the ambient reef water. This coral DOM selected for oligotrophic microbes of the *Alphaproteobacteria* and exerted only a small effect on the bacterioplankton community composition.

Most carbohydrates exuded by corals (36) and macroalgae (37–39) belong to the HMW size fraction (i.e., glycoproteins and polysaccharides, respectively), which can be extracted from seawater with ultrafiltration (UF ;~1 nm pore size) (40). The HMW carbohydrates exuded by corals and macroalgae can thus be concentrated using UF and subsequently added to ambient seawater with minimal dilution of the bacterioplankton community while, at the same time, keeping DOM concentrations within a natural range. The majority of studies investigating the effects of primary producer exudates on bacterioplankton apply dilution culture experiments (41), where prefiltered exudate-enriched water is inoculated with unfiltered ambient reef water resulting in a reduction of microbial cell abundance by at least 40% (11, 22, 28, 42). This approach alleviates density-dependent effects and allows to determine exponential growth rates as a measure of substrate quality. However, dilution also influences the community composition of bacterioplankton (43, 44) and affects competitive outcomes (45, 46).

Differences in the amount of exuded DOM between corals and macroalgae in previous studies further resulted in higher starting concentrations for macroalgae compared to coral DOM (22, 42), which makes it difficult to untangle the effects of DOM concentration (i.e., quantity) and DOM composition (i.e., quality) on bacterioplankton communities. Thus, the question remains whether coral and macroalgae exudates, added at similar concentrations, also differentially influence an undiluted microbial community.

All in all, interactions of coral- and macroalgae-derived DOM with bacterioplankton received much attention over the last decades (47) due to increasing macroalgae dominance, especially in the Caribbean (10), and global evidence that algae DOM supports reef microbialization (17). Nevertheless, it remains unclear how much of the previously observed effects on bacterioplankton were due to differences in composition vs concentration of coral- vs macroalgae-derived DOM (22). Finally, no study has yet assessed how the most abundant part of exuded carbohydrates, the HMW fraction, affects undiluted coral reef bacterioplankton communities.

To address these knowledge gaps, we investigated the exudation of HMW carbohydrates by corals and macroalgae, and how this particular DOM size fraction influences bacterioplankton communities from a Caribbean reef. We hypothesized that (i) coral and macroalgae HMW DOM differ in carbohydrate compositions and (ii) that these exudates enrich different taxa of the bacterioplankton community relative to seawater controls, with algae exudates exerting a stronger effect. We conducted a two-part experiment where we first incubated four hard coral species, two brown macroalgae genera, and seawater controls in aquaria to collect the exudates (see experimental design in Fig. 1). Subsequently, we concentrated HMW exudates from the incubation water and analyzed the monosaccharide composition of hydrolyzable carbohydrates. Finally, we added the concentrated HMW DOM to ambient seawater in 4-day dark incubations to elucidate effects on the growth and community composition of the heterotrophic bacterioplankton community to coral and macroalgae HMW DOM. By investigating bacterioplankton dynamics from a macroalgae-dominated reef in response

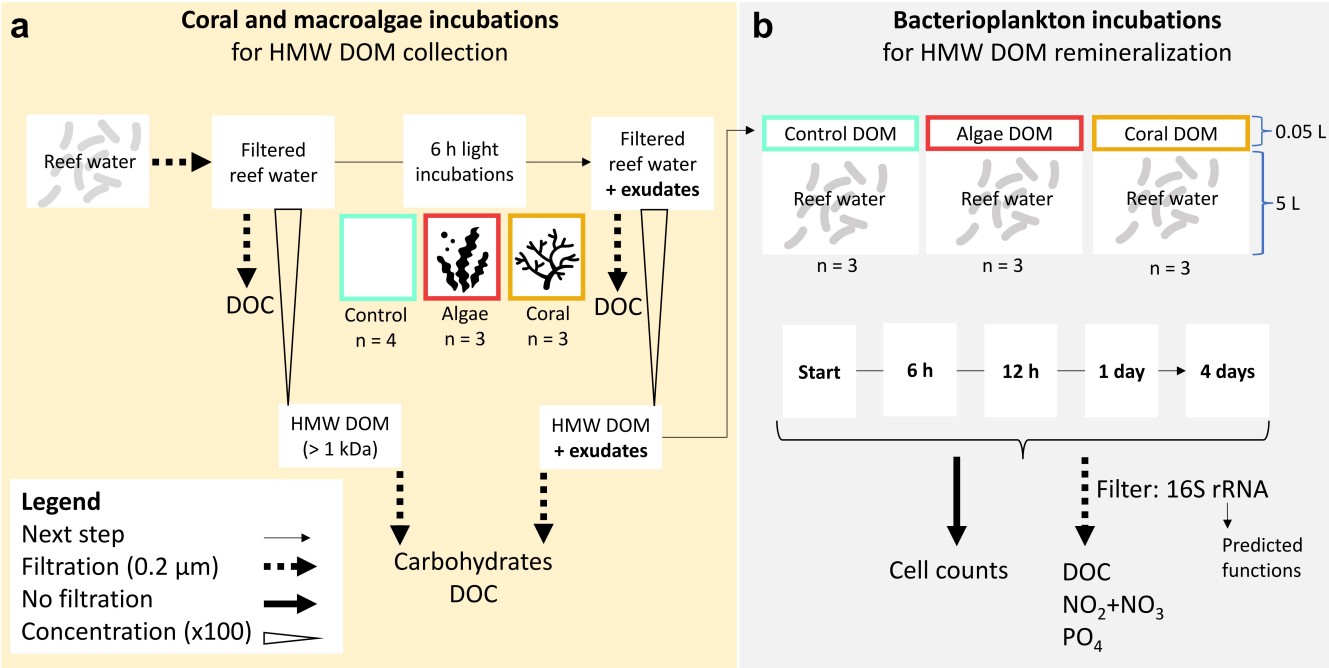

**FIG 1** Experimental design divided into (a) coral and macroalgae incubations for exudate collection and (b) bacterioplankton incubations for exudate remineralization by microbes from ambient reef water. DOC = dissolved organic carbon; HMW DOM = high molecular weight (>1,000 Dalton) dissolved organic matter. Concentrated HMW DOM (concentration factor of 100) enriched with exudates from coral and macroalgae incubations was diluted with ambient reef water in dark incubations (dilution factor of 100).

to primary producer-specific HMW DOM-carbohydrate compositions, our study may help to understand the functioning of changing reef communities from a microbial ecology perspective.

## MATERIALS AND METHODS

### Macroalgae and corals

The macroalgae *Dictyota* spp. and *Lobophora* spp. and three coral species (i.e., *Diploria labyrinthiformis*, *Meandrina meandrites*, and *Madracis auretenra*) were collected from the reef of Piscadera Bay, Curaçao (12.121, –68.970), at 8–10 m depth. Additionally, fragments of the coral *Acropora cervicornis* were provided by the coral restoration project Reef Renewal Curaçao and kept on a floating coral nursery (10 m depth) in front of the CARMABI research station until the experiment. All coral and algal species used here are abundant on Curaçaoan reefs and wide-spread through the Caribbean region (48–51). Algae were placed into re-sealable plastic bags for transportation and carefully rinsed with fresh seawater to remove sediment or particles before placing them into the experimental tank. Coral colonies were carefully cleaned with toothbrushes to remove encrusting sponges from the underside of colonies and kept in the floating nursery to heal any small injuries for 1 week. After retrieving macroalgae and hard corals from the reef/nursery, the organisms were kept submerged in a flow-through aquarium until the start of the incubations on the same day. Following the completion of experiments in the present study, coral colonies were placed back on the reef or used for restoration activities (i.e., *A. cervicornis*). All collections and experimental work were carried out under the research permit (#2012/48584) issued by the Curaçaoan Ministry of Health, Environment and Nature (GMN) to the CARMABI foundation. Research only included animals of lower taxonomic ranks (i.e., Cnidaria) and macroalgae, which do not require approval by an ethics committee according to German § 5 TierSchG (4 July 2013) and the European Directive 2010/63/EU (22 September 2010).

### Coral and macroalgae incubations for exudate collection

To collect coral and macroalgae exudates, as well as DOM from unamended seawater controls, 6-hour incubations were conducted in a glass tank (22 L, rinsed with 10% hydrochloric acid, HCl) with seawater, which was filtered to remove particles and planktonic organisms (see details below and experimental design in Fig. 1a). Corals or macroalgae were placed into the tank with sterile gloves (nitrile, powder free), covering about one-third (corals) or half (algae) of the aquarium bottom (see photographs in Fig. S1), which approximates the relative benthic cover at, respectively, coral- or algae-dominated reefs in the Caribbean (52). The same experimental setup was used for all incubations (three of each treatment and four seawater controls), which were done on different days between 8 and 23 November 2021. Seawater was constantly pumped from the reef (5 m depth) at Piscadera Bay to the aquarium facilities. One day prior to the experiment (i.e., control, algae, or coral incubations), seawater was directly collected from the inlet pipe and run over filters with decreasing pore sizes to remove particles (50, 20, 5, and 0.5 µm) and microbes (0.2 µm, 12 cm diameter, polycarbonate, pre-flushed with 1 L seawater) for 4 hours. Filtered seawater was collected and stored in closed high-density polyethylene (HDPE) buckets (rinsed with 10% HCl, ultrapure water [UW], and filtered seawater) in a dark, air-conditioned room (29°C) until the next day. The incubation tank was placed in a flow-through water bath with fresh seawater to keep temperatures close to *in situ* conditions (measured every 5 minutes with HOBO Pendant; Table 1). Salinity remained stable throughout the incubations (Table 1). A recirculating pump (UW-rinsed) was used to provide water movement in the incubation tank, and the top of the tank was left open to allow gas exchange. Oxygen concentrations remained constant in controls throughout the incubations but increased for both treatments during the incubations to 195% air saturation (Table 1). These oxygen concentrations can be observed in coral

**TABLE 1** Environmental parameters of incubations with seawater controls, corals, and macroalgae. Ambient values measured on the reef at 10 m depth (temperature, light) or at the start of the incubations (salinity, oxygen: $n = 10$)[a]

| Variable | Ambient | Control | Coral | Algae |
|---|---|---|---|---|
| Temperature (°C) | 29.1 ± 0.1 | 29.0 ± 0.3 | 29.0 ± 0.2 | 29.0 ± 0.1 |
| Light (PAR, μmol photons m$^{-2}$ s$^{-1}$) | NA[b] | 448 ± 6 | 448 ± 6 | 448 ± 6 |
| Salinity (‰) | 36.6 ± 0.2 | 36.7 ± 0.3 | 36.7 ± 0.1 | 36.9 ± 0.1 |
| Oxygen (μmol L$^{-1}$ / % air saturation) | 196 ± 4 / 115 ± 2 | 196 ± 4 / 115 ± 3 | 332 ± 23 / 195 ± 14 | 332 ± 23 / 195 ± 14 |
| Replicates ($n$) | 3/10 | 4 | 3 | 3 |

[a]Values are presented as means with SDs. Air saturation of 100% corresponds to 170 μmol $O_2$ L$^{-1}$ (or 6.8 mg L$^{-1}$). Temperature was measured every 5 minutes throughout the incubations, and salinity and oxygen were measured at the start (ambient) and end of the incubations. The light was measured once on 9 November 2021 for the experimental setup which was constant during the whole experiment.
[b]NA, not available.

reefs in diffusive boundary layers, at the coral-algal interface, and on the reef scale (53). Artificial light was provided throughout the incubations (CoralCare, Philips Lighting, 190-W-LED fixture). Light intensities for the experimental setup were measured on 9 November 2021 in photosynthetically active radiation (PAR; Table 1) and were within the range of light intensities on the reef at 10 m depth in November (i.e. up to ~500 μmol photons m$^{-2}$ s$^{-1}$ [13]).

## Concentration of high molecular weight dissolved organic matter

Water samples (~20 L) from the 6-hour aquarium incubations (start and end) were collected into HDPE buckets with a silicone tube (both 10% HCl and sample-rinsed). Corals and algae were removed with sterile gloves (nitrile, powder free) before air exposure could lead to a stress response which may affect the incubation water. From the bucket, water was filtered with pre-flushed (200 mL) polyethersulfone filters (0.2 μm pore size, Millipore Sterivex) for 2 hours using a peristaltic pump (Masterflex L/S) and sample-rinsed platinum-cured silicone tubes (Masterflex, 96410-15). Dissolved nutrient samples (i.e., DOC, inorganic nutrients) were collected from the filtrate. All remaining water was concentrated with tangential flow-filtration (TFF) by a factor of 100 and a molecular weight cutoff of 1,000 Daltons (UFP-1-C-5, Cytiva). The TFF cartridge was cleaned according to the manufacturer manual, with 0.5 M NaOH and ultrapure water. Trans-membrane pressure was set to 2–3 bar during sample concentrations. From the concentrated water (i.e., 14 L concentrated to 140 mL), samples for dissolved organic carbon (DOC) measurements were collected (i.e., HMW DOC). The remaining HMW concentrate was stored at −20°C for the microbial degradation experiment and carbohydrate analyses.

## Bacterioplankton incubations for exudate remineralization

To test the effects of HMW exudates from macroalgae and corals on bacterioplankton communities in reef water, we conducted 4-day dark incubations (see the experimental design in Fig. 1b). Fresh seawater was collected from the reef of Piscadera Bay on 1 December 2021 between 9:30 and 10:45 am at a depth of 6.8 m, 0.5 m above the benthos, with low current. Water was collected in four 20 L polyethylene terephthalate bottles (10% HCl and seawater rinsed) while facing the opening of the bottles away from the diver and toward the current. We decided to use unfiltered seawater (i.e., without removing potential grazers) to include larger aggregates and/or colloidal material, which are important hotspots of planktonic microbial diversity on coral reefs (54–56) to assess the effects of HMW DOM on bacterioplankton communities under natural conditions. Furthermore, prefiltration reduces grazing without concomitantly reducing viral lysis which may influence the microbial community composition (57). Water bottles were stored at 29°C in the dark for ~7 hours until the start of the experiment. Equal parts of every bottle were used for the nine incubation containers (three per treatment or

control, 5 L, polypropylene [PP]) to ensure equal water quality and microbial community compositions. HMW concentrates from coral and macroalgae incubations and seawater controls were thawed and pooled by treatment (i.e., concentrates from three replicate exudation incubations per treatment were mixed). Subsequently, 50 mL of the respective concentrate was added to 5 L of fresh seawater. All incubation containers were placed in dark Styrofoam boxes where HOBO loggers recorded light intensities (always 0 Lux) and temperatures (26.7°C ± 0.3°C SD). Samples for all parameters were collected at five timepoints: shortly after the start of the incubations (ca. 30 minutes after exudate addition, 1.3 L), after 6 and 12 hours (0.3 L each), and after 1 and 4 days (1.3 L each). Samples for DOC and inorganic nutrients were filtered with a peristaltic pump at a flow rate of 26 mL min$^{-1}$ (Masterflex L/S) through pre-flushed (200 mL) polyethersulfone filters (0.2 µm pore size, Millipore Sterivex) attached to sample-rinsed platinum-cured silicone tubes (Masterflex L/S Precision Pump Tubing, Tygon). Samples were collected from the bottom of each incubation container after gentle shaking.

## Coral mucus and macroalgae tissue extraction

Data on coral mucus carbohydrate compositions of *A. cervicornis*, *D. labyrinthiformis*, and *M. meandrites* were already reported in a previous study (58), where coral mucus collection and analyses are described in detail. Briefly, on the day after the coral and algae exudation experiment, coral colonies were exposed to air for 3 minutes. Dripping mucus was collected into sterile vials for 2 minutes after disposing mucus for the first minute, as done in previous studies (59) and stored at −20°C until analysis of carbohydrate compositions. Macroalgae from the experimental tanks were rinsed with UW and then oven-dried (40°C, 48 hours) and stored in sterile PP tubes (Falcon, 50 mL, Thermo Fisher Scientific) at room temperature in the dark. Dried algae tissue was pulverized with a mortar and pestle, and alcohol insoluble residue (AIR) and subsequently the water-soluble fraction were prepared as described by Vidal-Melgosa et al. (60). Dried powder was suspended in pure ethanol (volume ratio 6:1 of solvent:pellet), vortexed, rotated for 10 minutes, and then centrifuged (21,100 × *g* for 15 minutes at 15°C). The pellet was then washed in a 1:1 chloroform:methanol solution until the supernatant was clear, followed by a wash with pure acetone, and air-drying at room temperature. Subsequently, 10 mg of this AIR-washed powder was re-suspended in 300 µL of autoclaved UW, shaken for 2 hours at 15°C, centrifuged (6,000 × *g* for 15 minutes at 15°C), and the supernatant collected. The supernatant containing the water-soluble fraction was stored at −20°C until carbohydrate analysis.

## Carbohydrate analysis

For carbohydrate analysis of HMW DOM, 200 µL of 2 M HCl were added to 200 µL of samples and acid hydrolyzed at 100°C for 24 hours in pre-combusted (400°C, 4 hours) glass ampules. After hydrolysis, samples were dried down with an acid-resistant vacuum concentrator (Martin Christ Gefriertrocknungsanlagen GmbH, Germany) together with calibration standards which were prepared in 1 M HCl. Standards and samples were then resuspended in 200 µL UW and transferred to sterile high performance anion exchange chromatography (HPAEC) vials for measurement. Coral mucus samples and algae tissue extracts were hydrolyzed as described above and then diluted with UW by a factor of 100 (coral mucus) and 50 (algae tissue extracts), and centrifuged (21,100 × *g* for 15 minutes at 15°C), and 100 µL of the top layer was transferred to HPAEC vials for measurement with calibration standards. Monosaccharide concentrations of hydrolyzed carbohydrates were measured on a high-performance anion exchange chromatography (Dionex ICS-5000$^{+}$ system) with CarboPac PA10 guard column (2 × 50 mm) and analytical column (2 × 250 mm, Thermo Fisher Scientific), coupled with pulsed amperometric detection (HPAEC-PAD), as previously described (60, 61). Separation of neutral and amino sugars was achieved with an isocratic flow of 18 mM NaOH, followed by a separation of acidic monosaccharides with a gradient reaching up to 200 mM NaCH$_3$COO. Concentrations were calculated from peak areas of six co-measured standard mixes with concentrations

ranging from 10 to 1,000 µg L$^{-1}$ per monosaccharide using Chromeleon (Thermo Fisher Scientific).

## Dissolved organic carbon and inorganic nutrient analysis

DOC samples (20 mL) were collected in pre-combusted (4 hours at 450°C), sample-rinsed glass vials closed with teflon-lined lids (10% HCl acid washed and sample-rinsed), acidified with 5 drops of 12 M HCl to a pH <2, and stored at 4°C until measurement. Concentrations of DOC were measured with high-temperature catalytic oxidation (TOC-L CPN, Shimadzu), calibrated with a standard curve of potassium hydrogen phthalate (0; 25; 50; 100; 200; 400 µmol C L$^-$). Every sample was injected 5–7 times, resulting in an analytical variation of 2.3%. Measurement accuracy was tested by including consensus reference material (CRM; Batch 21, Lot: 04-21, DOC: 44.7 µmol L$^{-1}$ ± 0.8 SD, D. A. Hansell, University of Miami) into every measurement run, which was on average 8% below the reported concentration. Two samples (i.e., algae 2 and coral 2) had to be collected ~1.5 hours after the other samples due to logistical constraints, affecting DOC concentrations which were thus excluded from analyses.

Dissolved inorganic nutrient samples (5 mL) were collected in sample-rinsed HDPE vials (Midi-Vial, PerkinElmer; Waltham, MA, USA) and stored at −20°C until measurement on a Gas Segmented Continuous Flow Analyzer (QuAAtro / TRAACS, SEAL Analytical). Calibration standards were prepared in low-nutrient seawater with the same ionic strength as analyzed samples (salinity ~35‰). For the analysis of nitrite ($NO_2$) + nitrate ($NO_3$), $NO_3$ was first reduced to $NO_2$ at pH 7.5 by a copperized cadmium column. The resulting $NO_2$ was then turned into a pink complex through the addition of sulfanylamide and naphthyl ethylene-diamine and measured at 550 nm (62). Phosphate ($PO_4$) was first turned into a yellow phosphate-molybdenum complex at pH 1.0 with potassium antimonyl tartrate acting as a catalyst. The complex was then reduced into a blue molybdophosphate-complex with ascorbic acid and measured at 889 nm (63). Two samples from dark incubations had to be excluded (i.e., control 2, Start; coral 1, 1 day) due to high salinity (an artifact from overflowing sample vials while freezing).

## Microbial cell counts

Samples for microbial cell counts (1 mL) were taken with sterile pipette tips directly from the incubation containers, immediately fixed with 20 µL glutaraldehyde (15 min at 4°C) and stored at −80°C until analysis with flow cytometry (FACSCalibur, BD Biosciences, New Jersey, USA). Samples were 1:5 diluted with pre-filtered (0.2 µm) Tris-EDTA (TE) buffer, stained with 2% SYBR Green, and stored in the dark for >15 minutes prior to measurements. The output was analyzed with FCS Express 5 (DeNovo), and values from blanks (TE only) were subtracted from the results.

## DNA extraction

DNA of bacterioplankton communities was extracted from frozen (−20°C) polyethersulfone filters (0.2 µm pore size, Millipore Sterivex) which were used for nutrient sample collection (filtered volume: 0.2–1.2 L) as described in Haas et al. (64). All extractions were done with sterile utensils in a UV cabinet (UVT-S-AR, BioSan). We used the NucleaoSpin Tissue (Macherey-Nalgel, 250 preps) DNA extraction kit following manufacturer instructions with some adjustments. Filters were thawed at room temperature, air-dried with a Luer lock syringe, and closed on one end. Subsequently, 410 µL of proteinase K (50 µL) and buffer T1 (360 µL) solution was pipetted into the Luer lock opening of the filter, which was then closed and rotated overnight (~18 hours) at 55°C in a hybridization oven (Compact Line OV4, Biometra). On the next day, 400 µL of Buffer B3 was added to the filters, and filters were rotated at 70°C for 15 min. Subsequently, the liquid (500–750 µL) was removed from the filters with 3 mL Luer lock syringes and placed into a microtube. Samples were diluted with pure ethanol in a volume ratio of 1:2 ethanol:sample and vortexed. The solution was then added to the column and centrifuged for 1 min

at 11,000 × *g*. After washing the column with 500 µL of BW and 600 µL of B5 buffers, the silica membrane containing the extracted DNA was dried for 2 min at 11,000 × *g*. Pure DNA was then eluted into a microtube with 100 µL BE buffer which was incubated for 10 min at room temperature and spun down for 1 min at 11,000 × *g*. Extracted DNA was stored at −20°C and then transported on dry ice to the University of Hawaiʻi at Mānoa for amplification.

## Amplicon library sequencing and bioinformatics

The V4 region of the small subunit (16S) rRNA gene was amplified from each genomic DNA template using forward-indexed primers 515F-Parada: GTGYCAGCMGCCGCGGTAA and 806R-Apprill: GGACTACNVGGGTWTCTAAT (65) following the protocols of the Earth Microbiome Project (66), with the following minor changes: triplicate reactions were not pooled, and 1 µL was used of genomic template. Amplicons were cleaned and normalized to equimolar concentrations using the SequelPrep kit (Thermo Fisher Scientific) and then pooled and sequenced on the Illumina MiSeq platform (600 cycle V3 chemistry). Sequences were demultiplexed using a custom probabilistic script, and microbiome profiling was implemented in the MetaFlow|mics pipeline (67) following the specific settings presented by Jani et al. (68): sequences were assembled, denoised, and quality trimmed using DADA2 (69), globally aligned with mothur (70) to the Silva (v132) global SSU rRNA alignment database (71), bayesian consensus classified (70%) to the Genus level (72), and clustered into 99% sequence identity operational taxonomic units using vsearch (73) refined to reduce intragenomic taxonomic splitting errors using LULU (74). Sequences that could not be classified at the Domain level or were classified as chloroplast or mitochondrial 16S were discarded, and sequencing depth was subsequently standardized to 10,000 random reads per sample. All bioinformatics were deployed on the C-MAIKI gateway (75) at the University of Hawaiʻi at Mānoa by the Center for Microbial Oceanography: Research and Education.

We used MicFunPred (76) to predict metabolic pathway abundance from 16S rRNA data. The tool normalized our operational taxonomic unit (OTU) abundance table to library size, assigned taxonomy down to genus level, and then used the MetaCyc database to predict core gene contents. The output is a product of normalized taxonomic abundance and core gene content. We acknowledge the limitations of using 16S rRNA data to infer metabolic functions. First, the accuracy of predicting metabolic functions based on 16S rRNA amplicon sequencing depends on the quality and size of the reference database and may be limited for rare pathways and specific environments (76, 77). Second, pathways specific to certain strains are not included because 16S rRNA amplicons do not allow identification down to the microbial strain (77). MicFunPred addresses the second limitation, as it predicts the functional potential based on a set of core genes (i.e., genes present in ≥50% of genomes in the genus), which minimizes the false-positive results and uses MinPath to estimate the minimal set of pathways for pathway prediction (76).

## Statistical analyses

All statistical analyses were conducted using R Studio (R version 4.3.0, R Studio version 2023.06.2). Before conducting parametric tests, we checked for outliers (rstatix package), normal distribution (Shapiro-Wilk test), and homogeneity of variance (Levene test) across and within groups. The effect of treatments on DOC and carbohydrate concentrations in light incubations was tested with one-way analysis of variance (ANOVA) and Tukey's honestly significant difference (HSD) test for *post hoc* analyses. Concentration data of monosaccharides were log($x$ + 1) transformed, while mole % data were arcsine transformed. We used false discovery rate (fdr) correction to control the family-wise error rate across different monosaccharides. Hierarchical cluster analysis of HMW exudate-, coral mucus-, and algae tissue composition was conducted using Euclidean distance and the R package ComplexHeatmap (78). Simprof analysis (clustsig package) was used to test for significant differences between clusters. Differences of microbial cell densities

among timepoints (within-subjects factor) and treatments (between-subjects factor) were analyzed using two-way mixed model ANOVA (rstatix package) and pairwise $t$-tests with Bonferroni adjustment (i.e., to test for the effect of timepoints). For mixed-model ANOVA, we additionally tested for homogeneity of covariance (Box's M-test). Some outliers had to be excluded from DOC concentrations (i.e., $n = 2$ for algae and coral treatments) and inorganic nutrient concentrations (i.e., $n = 2$ for controls at the start and coral treatments after 1 day) and were thus analyzed with non-parametric tests (Kruskal-Wallis and Friedmann), followed by pairwise Dunn's tests with Bonferroni adjustment.

To compare microbial community composition and predicted metabolic pathway abundances among treatments, we used two-way permutational multivariate ANOVA (PERMANOVA) to test for the single and combined effects of time and treatment and five one-way PERMANOVAs with fdr-correction to test for treatment effects at every time point. For visualization of the microbial community composition, we used a non-metric multidimensional scaling (NMDS) plot (vegan package) with Bray-Curtis dissimilarity between square root transformed relative abundances of microbial genera. We used hierarchical cluster analysis of the Euclidean distance (ComplexHeatmap package) between $Z$-score scaled (across samples) relative abundances of microbial genera (>0.5% relative abundance) and predicted metabolic pathway abundance. For the timepoint with significant treatment effects on the microbial community composition (i.e., 4 days), permutational supervised classification random forest (RF) analysis was performed (rfPermute package, 5,000 trees, and 100 permutations), as done previously to assess differences in microbial community compositions between pre-defined groups (15, 79). Diagnostic plots confirmed that enough trees were grown. In addition, we used differential expression analysis (DESeq2 package [80]) to contrast microbial community composition and metabolic pathway abundance of both treatments with control communities. A consensus approach of RF and DESeq2 analyses was used to identify genera that were most important for characterizing treatments (81). Metabolic pathways were annotated by pathway class (i.e., super pathway, MetaCyc database) and pathway type (i.e., sub-groups of super pathways), and the abundance of each pathway class throughout the experiment was analyzed with mixed model ANOVAs (fdr-corrected). For metabolic classes with significant interaction terms, five one-way ANOVAs were conducted (one at each timepoint, Bonferroni-adjusted) and fdr-corrected for multiple testing across pathway classes.

## RESULTS

### Dissolved organic carbon and carbohydrate exudation by corals and macroalgae

At the end of the coral and macroalgae incubations, DOC concentrations were enriched by 13% and 11% compared to starting concentrations, respectively [ANOVA, $F_{(3,15)} = 10.5$, $P < 0.001$, $\eta^2_G$0.68, HSD test; Fig. 2a]. HMW DOC contributed 3.2% ± 0.7% (mean ± SD) to total DOC and was not significantly affected by treatments (Fig. 2b). Combined HMW carbohydrate concentrations were significantly enriched by 168% in macroalgae incubations compared to seawater controls and starting concentrations [ANOVA, $F_{(3,9)} = 6.5$, $P < 0.05$, $\eta^2_G$0.68, HSD test; Fig. 2c], while coral incubations were not significantly enriched (Fig. 2c). Percent carbohydrate content of HMW DOC in macroalgae and coral incubations was significantly enriched compared to controls and starting concentrations by 134% and 110%, respectively [ANOVA, $F_{(3,8)} = 13.0$, $P < 0.01$, $\eta^2_G$0.83, HSD test; Fig. 2d].

### Hydrolyzable monosaccharide composition of HMW coral and macroalgae exudates

In exudation incubations with macroalgae, concentrations of fucose, galactose, galacturonic acid, and rhamnose increased significantly by 388%, 252%, 128%, and 931%, respectively, compared to seawater controls and starting concentrations ($P < 0.01$,

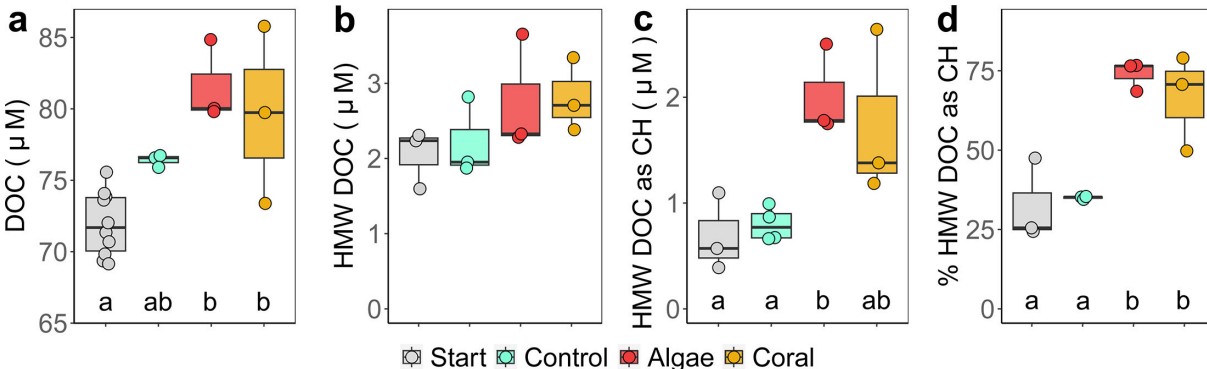

**FIG 2** Change during 6 hours of exudation incubations with corals and macroalgae in (a) DOC concentrations, (b) HMW DOC concentrations, (c) HMW carbohydrate concentrations (as carbon equivalents), and (d) percentage of HMW DOC as carbohydrates (CH). Values in panel c and d were calculated based on carbon contents of neutral-, amino-, and acidic monosaccharides measured after acid hydrolysis of HMW DOM (see Fig. 3). Boxes represent median ±95% confidence intervals, and points represent replicates. Different letters indicate significant differences between treatments (HSD test; $P < 0.05$). Raw data to this figure are available in Table S3.

HSD test, Fig. 3, all ANOVA results in Table 2). The mole percent contribution of fucose to total carbohydrates increased significantly compared to seawater controls and starting conditions, and mole percent contribution of rhamnose increased significantly compared to starting conditions ($P < 0.01$, HSD test, Fig. S2). All other monosaccharides were not significantly affected by macroalgae exudation.

In exudation incubations with corals, concentrations of arabinose, mannose, and glucosamine increased significantly by 812%, 225%, and 360% compared to seawater controls and starting concentrations ($P < 0.05$, HSD test; Fig. 3). The galactosamine concentration increased significantly by 125% compared to starting concentrations ($P < 0.05$, HSD test; Fig. 3). The mole percent contribution to total carbohydrates increased significantly compared to seawater controls and starting conditions for arabinose, mannose, and glucosamine ($P < 0.05$, HSD test; Fig. S2). All other monosaccharides were not significantly affected by coral exudation and none of the monosaccharides within the HMW DOM pool changed in seawater control incubations compared to starting concentrations (Fig. 3).

**TABLE 2** ANOVA results for differences in concentrations (Fig. 3) and mole percent compositions (Fig. S2) of HMW carbohydrates between treatments for all analyzed monosaccharides[a]

| Monosaccharide | Concentration | | | | Mole % | | | |
|---|---|---|---|---|---|---|---|---|
| | $F$ | $\eta^2$ | $P$ | $P$ (fdr) | $F$ | $\eta^2$ | $P$ | $P$ (fdr) |
| Fucose | 15.3 | 0.84 | <0.001 | **0.002** | 29.4 | 0.91 | < 0.001 | **<0.001** |
| Galactose | 17.0 | 0.85 | <0.001 | **0.002** | 5.4 | 0.64 | 0.021 | **0.028** |
| Galacturonic acid | 23.6 | 0.89 | <0.001 | **0.002** | 1.3 | 0.31 | 0.329 | n. s. |
| Rhamnose | 11.0 | 0.79 | 0.002 | **0.004** | 6.5 | 0.68 | 0.012 | **0.018** |
| Arabinose | 16.5 | 0.85 | <0.001 | **0.002** | 48.0 | 0.94 | < 0.001 | **<0.001** |
| Mannose | 6.8 | 0.69 | 0.011 | **0.019** | 7.6 | 0.72 | 0.008 | **0.016** |
| Glucosamine | 13.6 | 0.82 | 0.001 | **0.002** | 23.2 | 0.89 | < 0.001 | **<0.001** |
| Galactosamine | 5.7 | 0.65 | 0.018 | **0.027** | 6.8 | 0.69 | 0.011 | **0.018** |
| Xylose | 2.4 | 0.44 | 0.136 | n. s. | 3.7 | 0.55 | 0.054 | n. s. |
| Glucose | 1.4 | 0.33 | 0.293 | n. s. | 8.5 | 0.74 | 0.005 | **0.012** |
| Muramic acid | 0.4 | 0.12 | 0.755 | n. s. | 11.0 | 0.79 | 0.002 | **0.006** |
| Gluconic acid | 0.6 | 0.18 | 0.594 | n. s. | 0.8 | 0.21 | 0.517 | n. s. |

[a]Concentration data were log ($x$ + 1) transformed, while mole % data were arcsine transformed. DFn = 3, DFd = 9 for all tests. Bold values indicate significant (<0.05) fdr-corrected $P$ values. n. s. = not significant.

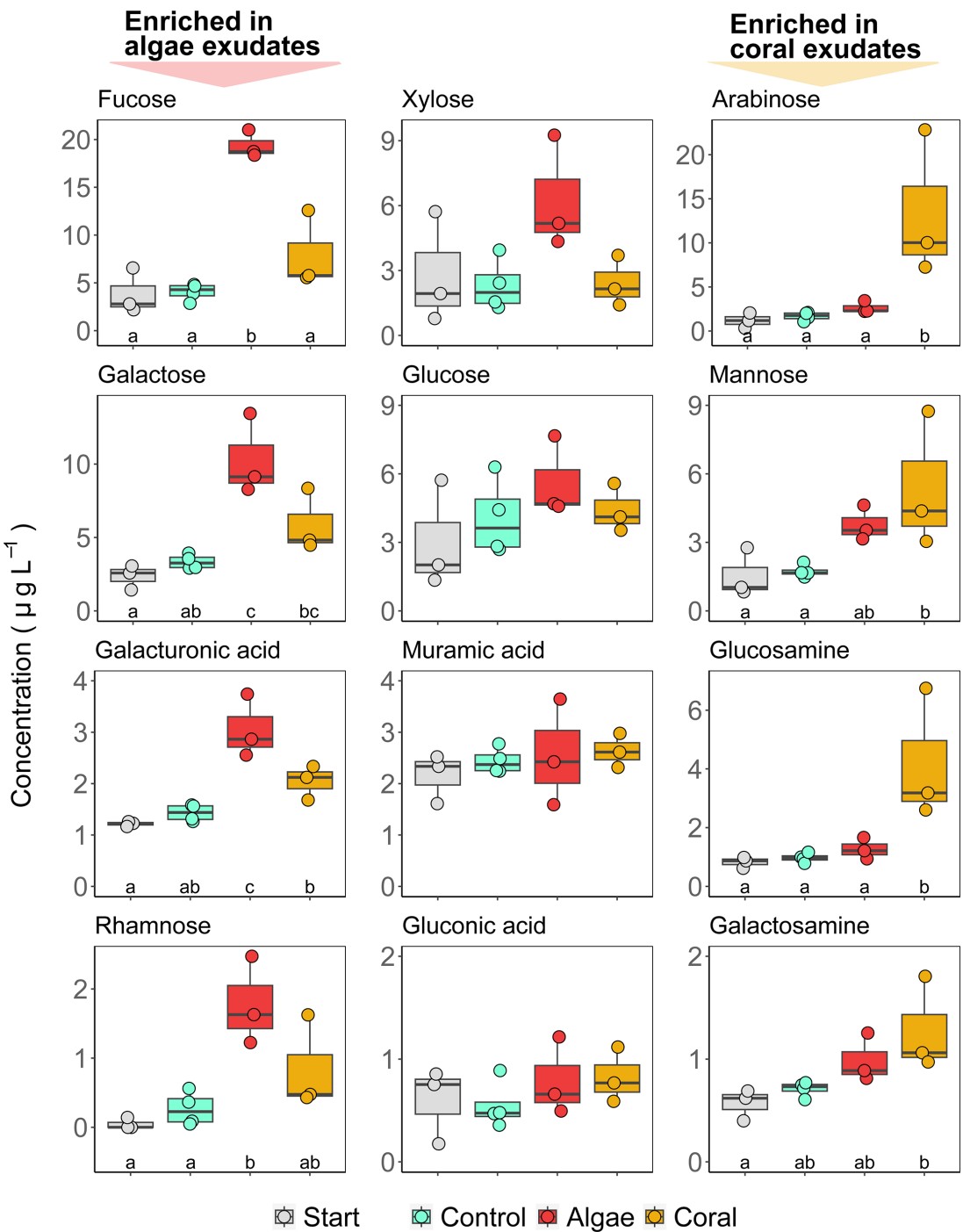

**FIG 3** Carbohydrate concentrations during incubations with corals and macroalgae. Boxes display concentrations of monosaccharides after hydrolysis of carbohydrates in the HMW fraction of DOM. Boxes represent median ±95% confidence intervals, and points represent replicates. Different letters indicate significant differences between treatments (HSD test; $P < 0.05$). Note differences in scale of the vertical axes. See ANOVA results in Table 2. Raw data to this figure are available in Table S4.

## Comparison of exudate compositions to coral mucus, macroalgae tissue, and ambient reef water

Coral exudate compositions clustered with coral mucus compositions of *A. cervicornis* ($n = 3$; SIMPROF, $P < 0.0001$; Fig. 4), which was characterized by high relative contents of arabinose (49% ± 4%; always stated as mean ± SD), glucosamine (24% ± 1%), and

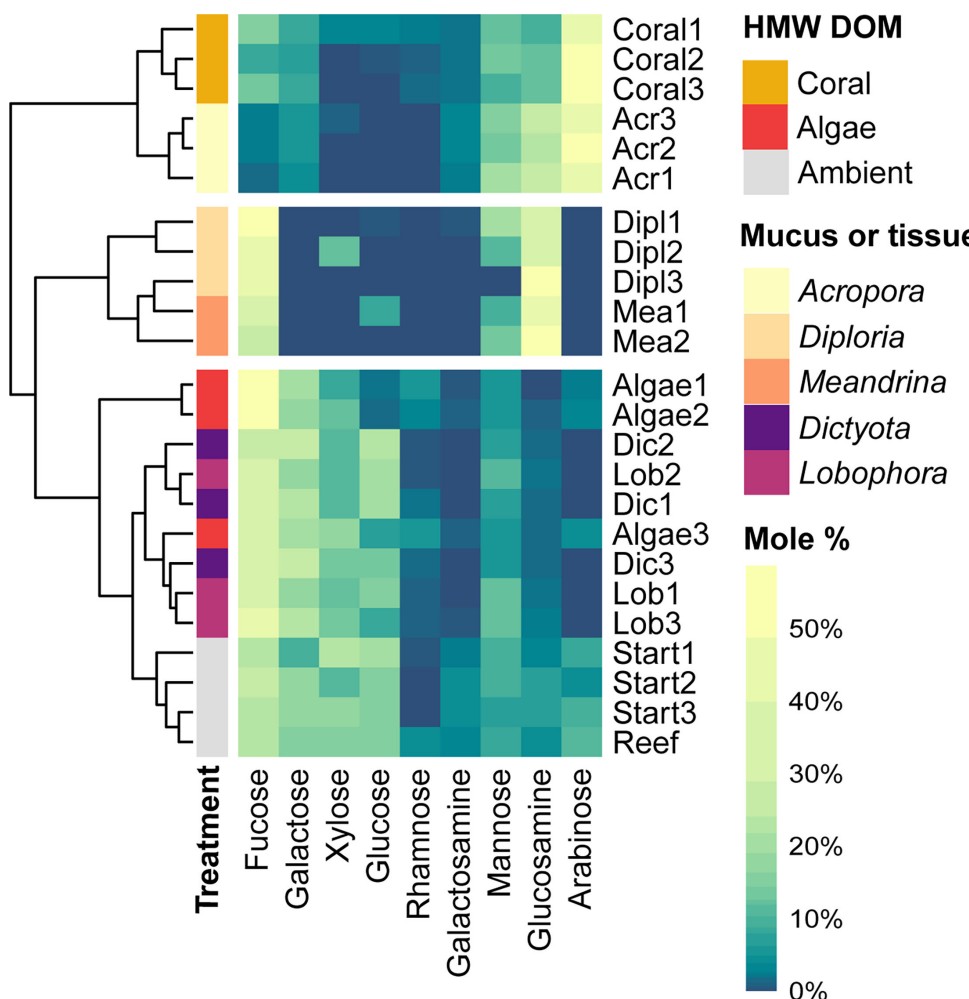

**FIG 4** Monosaccharide compositions of hydrolyzed coral and macroalgae HMW exudates (mole % composition of control-corrected fluxes) and ambient reef water HMW DOM in comparison to coral mucus and macroalgae tissue extracts. Coral mucus was collected from *A. cervicornis*, *D. labyrinthiformis*, and *M. meandrites*, and macroalgae tissue was extracted from dried *Lobophora* spp. and *Dictyota* spp. thalli. The dendrogram represents Euclidean distance between samples. Only neutral- and amino-sugars were used for the comparative cluster analysis. White spaces in heatmap separate clusters of samples which were significantly different in SIMPROF analysis (*P* < 0.0001). Raw data to this figure are available in Table S5.

mannose (17% ± 4%). Mucus compositions of *D. labyrinthiformis* (*n* = 3) and *M. meandrites* (*n* = 2) were significantly different from coral exudates (SIMPROF, *P* < 0.0001), with high relative contents of glucosamine (37% ± 13% and 52% ± 10%, respectively), fucose (48% ± 1% and 34% ± 7%, respectively), and mannose (11% ± 11% and 11% ± 3%, respectively). No monosaccharides could be detected in the hydrolyzed mucus of *M. auretenra*. Ambient reef water from the start of the exudation incubations ("Start," *n* = 3) and directly collected from the reef ("Reef," *n* = 1) clustered with algae tissue extracts and algae exudates (SIMPROF, *P* < 0.0001; Fig. 4) and was characterized by high relative contents of fucose (24% ± 3%), xylose (17% ± 5%), glucose (16% ± 2%), and galactose (15% ± 4%). The tissue extracts of *Dictyota* spp. (*n* = 3) and *Lobophora* spp. (*n* = 3) were not separated into individual clusters and were characterized by high relative contents of fucose (34% ± 4%), galactose (22% ± 4%), glucose (17% ± 6%), xylose (12% ± 1%), and mannose (9% ± 3%).

## Microbial cell densities and dissolved nutrient concentrations in bacterioplankton incubations

Microbes in bacterioplankton incubations grew in all treatments within the first day, with a mean growth rate of 401,837 ± 105,708 cells mL$^{-1}$ day$^{-1}$ which did not differ between treatments (ANOVA, $P = 0.6$). Microbial cell densities were similar between treatments and controls throughout the experiment (Fig. S3a). Only time had a significant effect on microbial cell densities [two-way mixed ANOVA: $F_{(2,9)} = 24.2$, $P < 0.001$, $\eta^2_G$G0.75]. Initial cell densities increased by 46%, 61%, and 22% after 12 hours, 1 day, and 4 days, respectively ($P < 0.05$, pairwise $t$-tests, Bonferroni adjusted, Fig. S3a). The estimated addition of HMW DOC for controls (i.e., the background material) was ~2.2 µM C, while the added HMW DOC from coral- and macroalgae incubations was ~2.8 µM C (i.e., mean HMW DOC concentrations shown in Fig. 2b diluted by a factor of 100, see experimental design in Fig. 1). The difference in DOC concentration between treatments and controls (~0.6 µM C) was thus likely insufficient to elicit a measurable differential growth response. Dissolved nutrient concentrations (Fig. S3b through d) averaged 88 ± 2 µM DOC, 0.5 ± 0.1 µM NO$_2$ + NO$_3$, and 0.013 ± 0.002 µM PO$_4$ at the start of the incubations, were not affected by treatments at any time throughout the experiment ($P > 0.05$, multiple Kruskal-Wallis tests), and declined over time [Friedmann tests; DOC: $\chi^2_{(4)} = 15.9$, $P < 0.01$; NO$_2$ + NO$_3$: $\chi^2_{(4)} = 15.4$, $P < 0.01$; PO$_4$: $\chi^2_{(4)} = 17.9$, $P < 0.01$]. After 4 days of dark incubation, DOC, NO$_2$ + NO$_3$, and PO$_4$ concentrations were reduced by 10%, 68%, and 26% compared to starting concentrations, respectively.

## Microbial community compositions in bacterioplankton incubations

The microbial community composition was significantly influenced by the interaction of time and treatment [PERMANOVA, $F_{(8)} = 2.0$, $R^2 = 0.03$, $P < 0.05$], with treatment only revealing significant effects after 4 days, explaining 52% of variation (Table 3; Fig. 5a). Both single factors were significant, with time and treatment alone explaining 88% and 2% of variation, respectively [PERMANOVA; Time: $F_{(4)} = 102.1$, $R^2 = 0.88$, $P = 0.001$; Treatment: $F_{(2)} = 4.6$, $R^2 = 0.02$, $P < 0.01$).

The initial microbial community was dominated by *Oxyphotobacteria* of the *Synechococcales* order and *Alphaproteobacteria* of the *SAR11* order (Fig. S4). After 1 day, *Oxyphotobacteria* declined in relative abundance and were mainly replaced by *Gammaproteobacteria* of the *Alteromonadales* order. The change from days 1 to 4 of the experiment was mainly due to increased relative abundance of *Alphaproteobacteria* of the *SAR11* and *Rhodobacterales* orders (Fig. S4).

Hierarchical cluster analysis of the genus-level microbial community (genera with a maximum in any one sample greater than 0.5% rel. abundance, *Z*-score) revealed a separate cluster of coral samples from controls and algae exudate enriched samples after 4 days (Fig. 5b), albeit clustering was not significant (SIMPROF: $P > 0.05$). To identify the most important genera for the classification of treatments, permutational RF classification models were applied. In the coral exudate treatment, 16 genera contributed significantly (PFpermute: $P < 0.05$) to the classification after 4 days (see green bars in Fig. 5c; Fig. S5). In contrast, only one genus was driving the classification of algae exudate-enriched communities. All coral exudate communities were successfully classified by the RF model, while algae exudate samples could not be distinguished from controls (Table S1). In coral exudate treatments, 17 genera were significantly different from controls (DESeq2: $P < 0.05$, fdr-corrected, Fig. 5c), while algae exudate treatments did not reveal any significant differences to controls. The consensus of both methods (RF and DESeq2) revealed eight genera (see black genus names in Fig. 5c) which characterized coral exudate treatments after 4 days.

Microbial genera significantly enriched in coral exudate treatments compared to controls (based on the consensus of RF and *DESeq2*) belonged to the *Gammaproteobacteria* (*Vibrionaceae* and *Thiotrichaceae*), *Bacteroidia* (*Flavobacteriales* and *Saprospiraceae*), *Alphaproteobacteria* (*Rhodobacteraceae*), and *Planctomycetes* (*Phycisphaeraceae*,

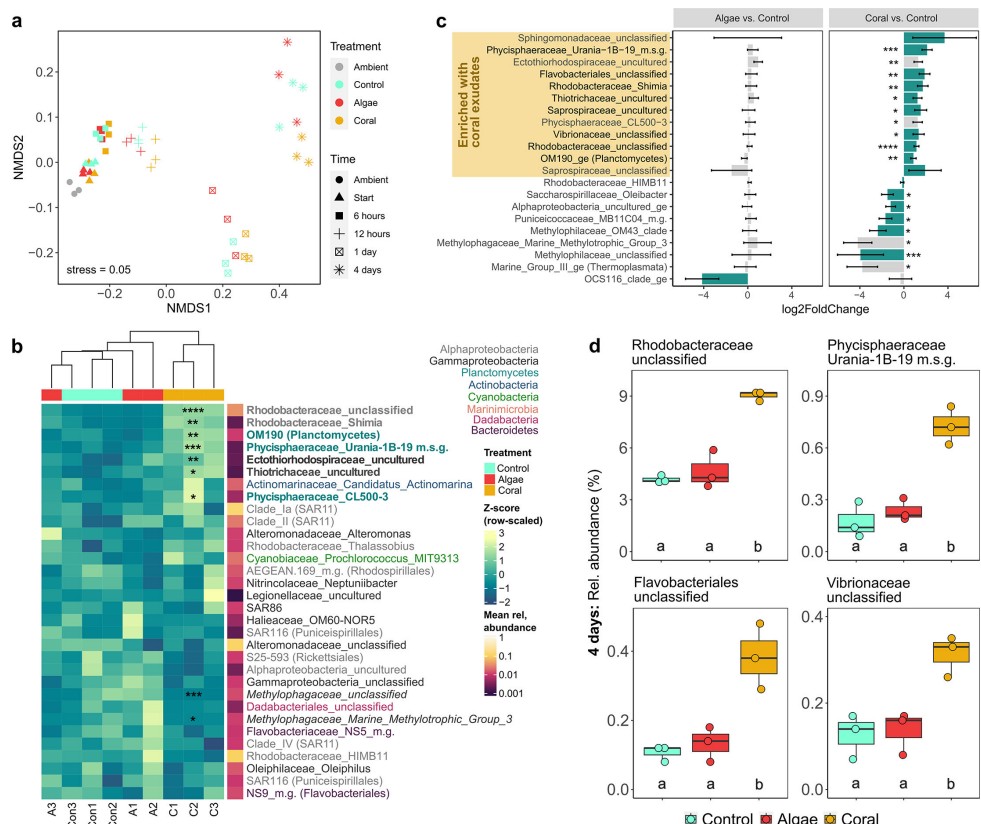

**FIG 5** Microbial community composition in dark incubations of reef water with addition of HMW DOM of macroalgae and coral exudates and seawater controls, (a) for all timepoints, and (b-d) after 4 days (final timepoint), when a significant treatment effect was observed ($P < 0.05$, PERMANOVA; Table 3). (a) NMDS of square root transformed genus-level microbial community compositions (Bray-Curtis dissimilarity matrix). (b) Row-scaled abundance (Z-score) of genera (>0.5% mean relative abundance in any one sample) in hierarchical clustering heatmap (Euclidean distance). Bold lineage names indicate an increase, and italicized names indicate a decrease with coral exudates compared to controls. (c) Log2 fold change of both treatments vs controls. Error bars represent SE (lfcSE of DESeq2 output). Green bars indicate genera that were significant ($P < 0.05$, importance score >5) in RF model for classifying algae or coral treatments. Black genera names indicate consensus of RF and DESeq2 analysis. (d) Relative abundance of four taxa which were significantly ($P < 0.05$) different in coral treatments compared to controls and algae treatments after 4 days of incubation. Asterisks (in panels b and c) indicate significant coral treatment effect vs controls (****$P < 0.0001$, *** $P < 0.001$, ** $P < 0.01$, and * $P < 0.5$, DESeq2, fdr-corrected). M.g. = marine group; m.s.g. = marine sediment group. Raw data to this figure are available online (10.5281/zenodo.13365525), and DESeq2 results (c) are available in Table S7.

**TABLE 3** PERMANOVA results of treatment effects on the microbial community composition (genus level) and predicted metabolic functions at five times during bacterioplankton incubations, using 999 permutations[a]

| Time | Microbial community composition | | | | Predicted metabolic functions (MicFunPred) | | | |
|---|---|---|---|---|---|---|---|---|
| | $F$ | $R^2$ | $P$ | $P$ (fdr) | $F$ | $R^2$ | $P$ | $P$ (fdr) |
| Start | 0.60 | 0.17 | 0.70 | 0.70 | 0.5 | 0.15 | 0.91 | 0.91 |
| 6 hours | 0.91 | 0.23 | 0.47 | 0.59 | 2.1 | 0.41 | 0.14 | 0.18 |
| 12 hours | 1.30 | 0.30 | 0.23 | 0.39 | 4.0 | 0.57 | 0.05 | 0.13 |
| 1 day | 3.39 | 0.53 | 0.04 | 0.10 | 2.2 | 0.42 | 0.12 | 0.18 |
| 4 days | 3.21 | 0.52 | 0.01 | 0.045 | 13.3 | 0.82 | 0.02 | 0.11 |

[a]Df = 2 for all tests, fdr = false discovery rate.

*OM190* class; see black genus names in Fig. 5c). The most abundant member of the coral-enriched microbial community was an unclassified genus of the *Rhodobacteraceae* (coral = 9.0%, control = 4.2%, and algae = 4.7%; Fig. 5b), followed by *Planktomycetes* of the uncultured class *OM190* (coral = 1.7%, control = 0.5%, and algae = 0.6%), and the genus *CL500-3* of the *Phycisphaeraceae* (coral = 1.2%, control = 0.9%, and algae = 0.8%). All other coral exudate-enriched genera were still rare (<1%).

## Predicted metabolic functions

Predicted metabolic functions of the microbial communities (using MicFunPred) were significantly affected by the interaction of time and treatment [PERMANOVA, $F_{(8)} = 3.5$, $R^2 = 0.02$, $P < 0.01$] with the strongest treatment effect after 4 days, albeit not significant after $P$ value correction (Table 3). Metabolic class abundance was significantly affected by the interaction of time and treatment for 8 out of 13 classes, and seven of these increased significantly in coral treatments compared to controls and algae treatments after 4 days (Table 4; Fig. 6a). Hierarchical cluster analysis of the $Z$-score adjusted pathway type abundance revealed a separate cluster of coral samples from all other samples after 4 days (SIMPROF; $P < 0.0001$; Fig. 6b).

Predicted pathways associated with energy metabolism in HMW coral DOM incubations increased by 28% ($P < 0.001$, pairwise $t$-tests, Bonferroni adjusted, Fig. 6a), with significant increases in 6 out of 10 pathway types, including glycolysis, Entner Duodoroff, and pentose phosphate pathways ($P < 0.001$, DESeq2, fdr-corrected, Fig. 6b). Predicted amino acid metabolism increased by 44% ($P < 0.001$), both in biosynthesis and degradation pathways, though not significantly for a specific pathway type. Predicted carbohydrate metabolism increased by 111% ($P < 0.0001$), with significant increases in 7 out of 12 pathway types, including both degradation and biosynthesis pathways ($P < 0.001$). Predicted fatty acid and lipid metabolism increased by 24% ($P < 0.05$), with a significant increase in fatty acid degradation (i.e., one out of nine pathway types, $P < 0.001$). Predicted pathways associated with secondary metabolites increased by 63% ($P < 0.01$), with significant increases in terpenoid and polyketide biosynthesis (i.e., two out of six pathway types, $P < 0.001$). Other predicted degradation pathways increased by 22% and other biosynthesis pathways by 9% (see Fig. S6 for more detail).

**TABLE 4** Statistical test results for treatment and time interaction effect (DFn = 8 and DFd = 24) of mixed model ANOVA, and treatment effect (DFn = 2 and DFd = 6) after 4 days of bacterioplankton incubation for all metabolic classes with significant interaction effects[a]

| Metabolic class | Mixed ANOVA: treatment × time | | | | ANOVA: treatment after 4 days | | | | % of total pathways | % coral effect size |
|---|---|---|---|---|---|---|---|---|---|---|
| | F | $\eta^2$ | P | P (fdr) | F | $\eta^2$ | P (BF) | P (fdr) | | |
| Energy metabolism | 7.3 | 0.68 | 0.007 | **0.015** | 59.8 | 0.95 | 0.001 | **0.001** | 14 | 28 |
| Amino acids | 7.6 | 0.70 | 0.000 | **0.000** | 83.2 | 0.97 | 0.000 | **0.001** | 12 | 44 |
| Carbohydrates | 6.2 | 0.65 | 0.016 | **0.026** | 640 | 0.99 | 0.000 | **0.000** | 6 | 111 |
| Fatty acids and lipids | 4.3 | 0.53 | 0.003 | **0.008** | 12.9 | 0.80 | 0.035 | **0.040** | 7 | 24 |
| Secondary metabolites | 8.7 | 0.73 | 0.000 | **0.000** | 26.8 | 0.90 | 0.005 | **0.007** | 4 | 63 |
| Other degradation | 11.8 | 0.78 | 0.000 | **0.000** | 27.2 | 0.90 | 0.005 | **0.007** | 9 | 22 |
| Other biosynthesis | 3.1 | 0.48 | 0.014 | **0.026** | 27.1 | 0.90 | 0.005 | **0.007** | 11 | 9 |
| Inorganic nutrients | 4.4 | 0.55 | 0.002 | **0.007** | 5.4 | 0.64 | 0.230 | 0.230 | 7 | – |
| Detoxification | 2.3 | 0.39 | 0.054 | 0.078 | – | – | – | – | 4 | – |
| Cofactors, carriers, and vitamins | 2.1 | 0.39 | 0.077 | 0.100 | – | – | – | – | 12 | – |
| Cell structure biosynthesis | 2.0 | 0.35 | 0.186 | 0.213 | – | – | – | – | 1 | – |
| Nucleosides and nucleotides | 2.0 | 0.38 | 0.197 | 0.213 | – | – | – | – | 11 | – |
| Pigments | 0.6 | 0.14 | 0.677 | 0.677 | – | – | – | – | 1 | – |

[a]$P$ (BF) = Bonferroni-adjusted $P$-values for multiple one-way ANOVAs (i.e., one at each timepoint, only the result for 4 days is reported). % of total pathways = percent contribution of mean pathway abundance per class to the sum of all pathways after 4 days; % coral effect size = percent increase with coral exudates compared to controls after 4 days. Bold values indicate significant (<0.05) fdr-corrected $P$ values. A dash indicates when no further analysis was done.

## DISCUSSION

Our results (summarized in Fig. 7) revealed that brown macroalgae, as well as scleractinian corals, exude significant amounts of HMW carbohydrates (Fig. 2d) which differ in composition (Fig. 3). The compositional differences in exuded HMW DOM, added at

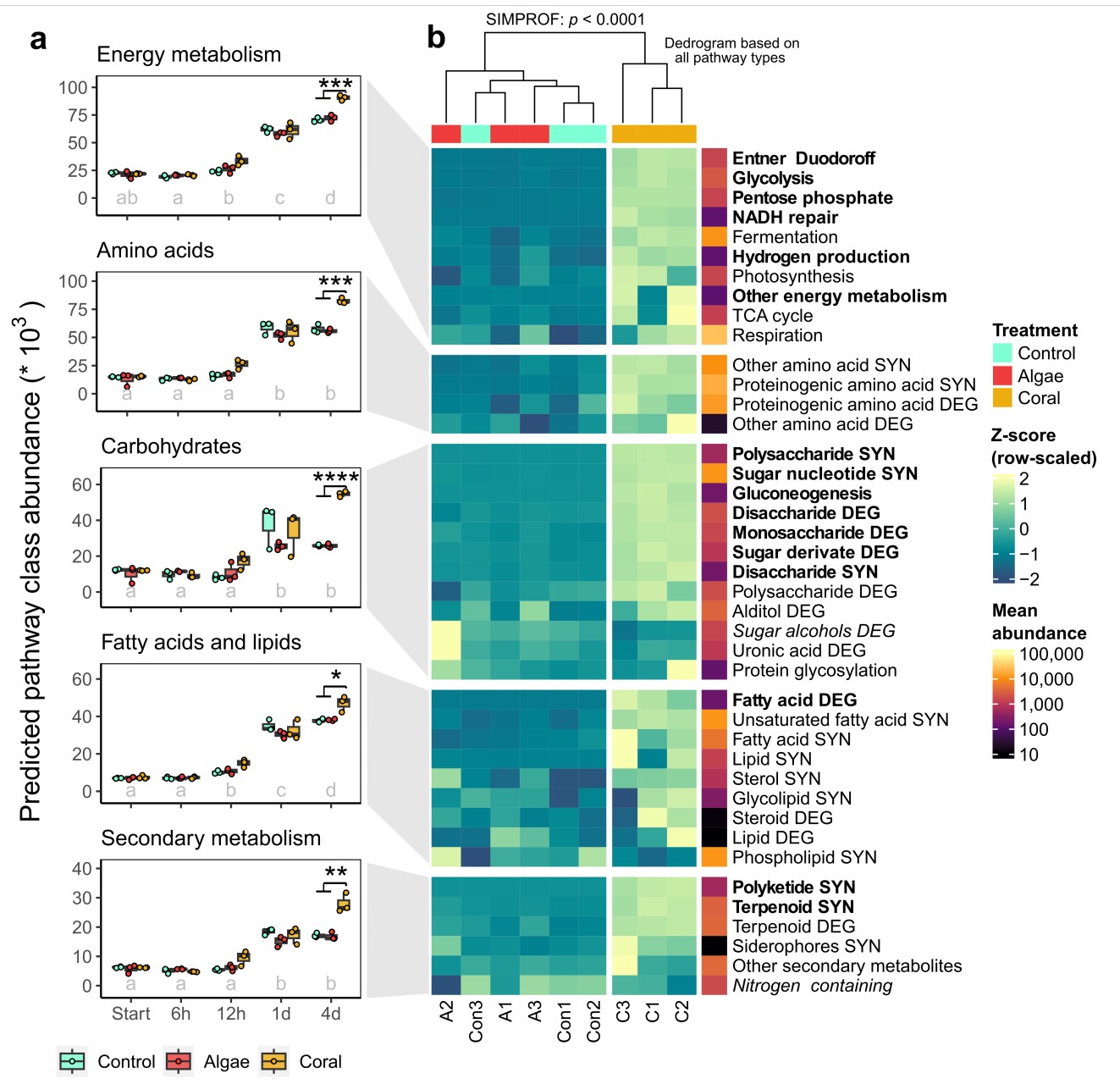

**FIG 6** (a) Development of predicted metabolic pathway class abundances during bacterioplankton incubations with HMW coral- and macroalgal-DOM and background HMW DOM (controls). Different letters indicate significant differences between times ($P < 0.05$, pairwise $t$-tests, Bonferroni adjusted). Asterisks indicate significant differences between treatments (*$P < 0.05$, ***$P < 0.001$, and ****$P < 0.0001$, pairwise $t$-tests, Bonferroni adjusted). Predictions are based on the MicFunPred analysis tool, using the MetaCyc database. (b) Predicted pathway abundance by type for the significantly enriched classes as $Z$-scores after 4 days of dark incubation (final timepoint). The dendrogram on the top represents the Euclidean distance between samples including all pathway types of all classes. SYN = biosynthesis and DEG = degradation. Bold pathway types in panel b indicate a significant increase in coral treatments compared to controls (fdr-corrected $P < 0.001$, log2 fold change > 0.5, DESeq2 on all pathway types), and italicized types a significant decrease. Raw data to this figure are available (a) in Table S8 and (b) at doi 10.5281/zenodo.13365525.

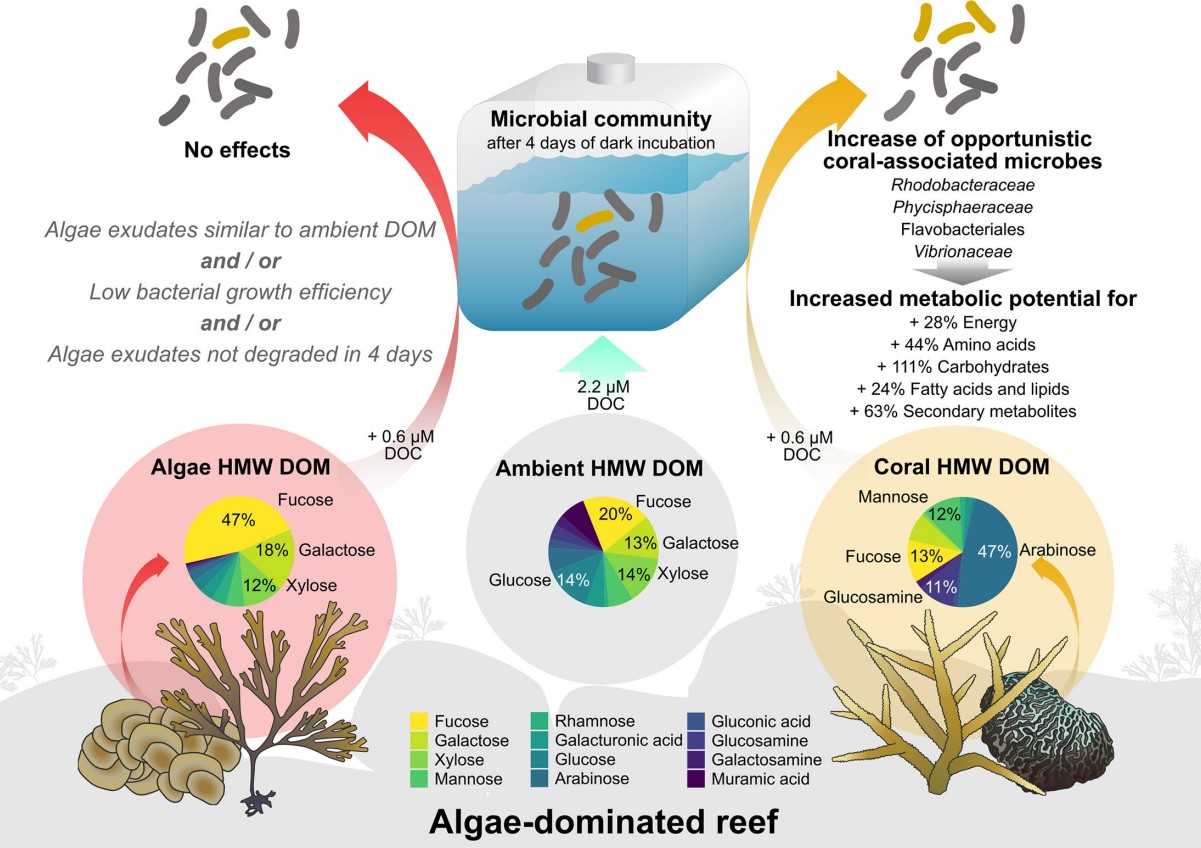

**FIG 7** Summary of HMW DOM compositions of coral and macroalgae exudates and microbial community responses. Pie charts display mean mole percent compositions of control-corrected fluxes or ambient seawater composition. Values >10% are shown in the figure. Gray italics indicate possible explanations for the lack of microbial response to algal HMW exudates. Icon attribution: Integration and Application Network (ian.umces.edu/media-library).

low concentrations (<1% of ambient DOC), had no effect on the overall bacterioplankton cell density or dissolved nutrient concentrations (Fig. S3). However, coral HMW DOM significantly affected the bacterioplankton community composition (Fig. 5) and increased the predicted potential for specific metabolic functions relative to controls (Fig. 6). In contrast, algae HMW DOM addition induced no significant differences compared to seawater controls.

## Coral exudates reflected HMW carbohydrate composition of coral mucus and enriched opportunistic coral-associated microbes

The HMW carbohydrates released by corals were enriched in arabinose, glucosamine, mannose, and galactosamine (Fig. 3) and mostly reflected the monosaccharide composition of hydrolyzed mucus from *A. cervicornis* (Fig. 4). The main carbohydrate-containing macromolecules in coral mucus are mucin glycoproteins (i.e., 0.5–50 mDa) (82), and carbohydrate side chains of mucins isolated from *Acropora formosa* were rich in arabinose, glucosamine, and mannose (83), supporting the presence of mucins in HMW coral-DOM. Furthermore, although the prediction of metabolic functions from 16S rRNA data is limited in accuracy (76, 77) (see Materials and Methods for details), the increase in predicted carbohydrate, amino acid, and fatty acid and lipid metabolism in bacterioplankton communities (Fig. 6) is consistent with degradation of coral mucus, which mainly consists of carbohydrates, proteins, and lipids (84, 85).

The minor addition of HMW coral DOM (~0.6 µM C; <1% of DOC) to an ambient bacterioplankton community significantly increased the relative abundance of several genera belonging to the *Alphaproteobacteria* (*Rhodobacterales*), *Planctomycetes*

(*Phycisphaerales*, *OM190* class), *Gammaproteobacteria* (*Vibrionales* and *Thiotrichales*), and *Bacteroidetes* (*Flavobacteriales* and *Saprospirales*; Fig. 5). However, the increase in microbial cell densities over time was similar to those observed in HMW macroalgae- and seawater control DOM incubations (Fig. S3). Bacterioplankton growth kinetics in dilution cultures usually include an exponential growth phase followed by a stationary phase (41), where the stationary abundance at least partially depends on the amount of substrate added (22, 86). As we only diluted the microbial community by ~1% and added low and similar amounts of substrate for treatments (2.8 µM DOC) and controls (2.2 µM DOC), we did not expect differences in final microbial cell abundances. *Rhodobacterales*, *Thiotrichales*, *Vibrionales*, *Flavobacteriales*, and *OM190* have been previously reported in coral mucus (87–91) and/or increased in seawater when coral mucus (92–94) or coral exudates (22, 28) were added (Table S2). *Phycisphaeraceae* were previously found in *Acropora hyacinthus* (95), deep-sea corals (96), and in association with cultured Symbiodiniaceae (97). Thus, HMW coral-DOM enriched bacterioplankton taxa are commonly associated with corals.

*Rhodobacterales*, *Vibrionales*, and *Flavobacteriales* were previously found to increase with macroalgae DOM addition and are considered to be opportunistic heterotrophic bacteria adapted to fast growth on DOM (22). Furthermore, they can increase in coral holobionts under stress (98, 99) and disease (100, 101) and were therefore suggested as indicators for poor reef health (102). Additionally, *Thiotrichaceae* (103) and *Phycisphaeraceae* (95, 104) can be associated with coral disease, and *OM190* increased in bacterioplankton during a marine heatwave (105). Thus, HMW coral DOM mostly selected for coral-associated microbes with the ability for opportunistic growth, which is also supported by the increase in predicted carbohydrate-, amino acid-, and fatty acid metabolism (Fig. 6) (106).

## Algae exudates reflected algae tissue and ambient reef water HMW carbohydrate composition and did not affect the bacterioplankton community composition

HMW carbohydrates released by the two brown macroalgae *Dictyota* spp. and *Lobophora* spp. were enriched in fucose, galactose, galacturonic acid, and rhamnose (Fig. 3) and were similar in composition to tissue extracts from both species (Fig. 4). Furthermore, the HMW carbohydrate composition of ambient reef water mostly reflected algae exudate compositions (Fig. 4 and 7). In contrast, the previously reported carbohydrate composition of ambient reef water from Moorea in the central Pacific was most similar to that of coral exudates (22). This suggests that the composition of HMW DOM in the reef may depend on the community of benthic primary producers fueling and shaping the local DOM pool (17, 107). Indeed, our study site is characterized by high algae (i.e., 17% macroalgae and 13% turf algae) and low coral (i.e., 7%) cover (108), with *Dictyota* and *Lobophora* being both highly abundant and releasing substantial amounts of DOM (12, 13, 109). On the other hand, the DOM pool of the rapidly flushed backreef of Moorea appears to be at least partly fueled by dense coral communities on the outer reefs (22, 110, 111).

Microbial communities growing on HMW macroalgae DOM did not reveal any differences to seawater control incubations, thus not confirming the hypothesis that algae HMW exudates exert stronger effects on bacterioplankton communities. However, our addition of DOM was different from previous studies in several ways. We enriched the microbial community exclusively with HMW DOM and thus removed any molecules <1 kDa from the exuded fraction. Freshly produced exudates can contain LMW DOM with high bioavailability like free monosaccharides and amino acids, which are taken up rapidly by microbes (112) and could have contributed to the microbial response in previous studies. Further studies using undiluted bacterioplankton communities in combination with a natural ratio of HMW and LMW coral and macroalgal DOM are required to fully unravel the control of DOM source (and thereby composition) on bacterioplankton communities. Furthermore, the DOC addition in the present study

(~2–3 µM DOC) is more than one order of magnitude lower compared to previous studies (~40–100 µM DOC [22, 42]). These comparable high DOC additions allowed for high DOC uptake rates to compensate for low bacterial growth efficiencies (i.e., increased respiration at the expense of biomass formation) of algal-DOM, resulting in higher microbial growth rates on algal- vs coral-DOM (19, 22, 42). In the present study, there were neither differences in initial DOC concentrations between coral and macroalgae exudates (Fig. 2a and b) nor differences in microbial growth rates (Fig. S3a). Our approach thereby allowed a decoupling of DOM concentration-dependent from DOM composition-dependent effects, which suggest that the increased DOC concentration component at least partly (excluding potential effects from LMW DOM) explained previously reported differences in bacterial growth rates between coral and algae DOM.

HMW macroalgae DOM could have also resisted microbial degradation throughout the 4-day incubations. Brown macroalgae can exude large quantities of fucoidan (38), a complex fucose-rich polysaccharide that forms an extracellular matrix and prevents desiccation of algal thalli (113, 114). Fucose was the most abundant monosaccharide in HMW DOM of the brown macroalgae in the present study, as well as for the brown algae *Turbinaria* on reefs in French Polynesia (22). Using monoclonal antibodies (see supplemental methods), we detected three epitopes present in sulfated fucan (BAM1, BAM3, and BAM4) in macroalgae tissue (Fig. S7a), which is consistent with previous findings of fucoidan in the tissue of both genera (113, 115). Additionally, relative proportions of fucose, galactose, xylose, and mannose were similar in fucoidan extracted from *Dictyota* spp. and *Lobophora* spp. compared to macroalgae exudates (Fig. S7b), suggesting that fucoidan contributed to macroalgae HMW DOM. Microbial degradation of brown algae fucoidans is energetically costly because it can require hundreds of different enzymes to degrade its complex structure (116). This could also explain why bacterioplankton growing on brown macroalgae exudates incorporate less carbon and have higher respiratory costs compared to green algae exudates (22, 42), as green algae do not contain fucoidan in their cell walls (117). Similarly, the high contribution of fucose, galactose, xylose, and mannose in ambient reef water (combined making up 54% of HMW carbohydrates, Fig. 4 and 7) suggests a considerable abundance of fucoidan in the HMW fraction of reef water, which supports the high resistance of brown macroalgae HMW DOM to microbial degradation.

## Ecological implications

Previous studies were mainly conducted on coral-dominated reefs and found that coral exudates support a diverse oligotrophic bacterioplankton community, while algae exudates promote opportunistic microbial taxa (22) and less energy-efficient nutrient cycling (17). A shift toward macroalgae dominance can thus reduce ecosystem productivity by enhancing microbial respiration (11, 118) and decrease the transfer of energy to higher trophic levels (17, 18). The strong effects observed here from a small addition of coral exudates on the bacterioplankton community composition support the energy-efficient transformation of coral exudates into microbial biomass (11, 17). However, the increase in mainly opportunistic microbial taxa with coral exudates seems to contradict previous results. A change in carbon substrates can act as a disturbance on microbial communities (119). Thus, the addition of coral exudates to a microbial community from a reef with macroalgae DOM dominating the local DOM pool could be considered a disturbance of the alternative stable state (i.e., the algae-dominance [120]). A common ecosystem response to stress is a shift toward opportunists which are less specialized but respond rapidly to perturbations by adapting to new environmental conditions (121, 122). We, therefore, propose that the increase of some opportunistic microbial taxa on reefs may not be a direct response to the exudates of a specific primary producer but rather to a disturbance in the form of a change in the availability of DOM producer-specific carbon substrates.

Brown macroalgae HMW DOM did not appear to support microbial growth, which could be explained by increased respiration instead of biomass production (not

measured here but shown previously [11, 17, 19]). An additional explanation could be that resistant HMW molecules form brown macroalgae such as fucoidan defied degradation during the 4 days of dark incubations. Previous studies revealed resistance of brown algae exudates to microbial degradation for up to 5 months, which leads to a net export of DOM from brown macroalgae beds (123–126). Water residence times of fringing reefs can range from hours to days (127, 128). Thus, brown algae exudates could be exported from coral reefs. Release of refractory DOM by brown macroalgae which replace corals on many reefs could be an additional pathway by which the transfer of energy to higher trophic levels declines on degraded reefs. This hypothesis could be tested by measuring fucose concentrations at gradients away from algae-dominated reefs, as fucose can function as a biomarker for brown algae origin (129).

## Effects of changing DOM compositions beyond coral reefs

Our results indicate that opportunistic microbial taxa increase in bacterioplankton communities of coral reefs following a change in the main DOM substrates (here induced by addition of coral DOM to macroalgae DOM-dominated ambient reef water; Fig. 7). This hypothesis is based on the *r*- and *K*-selection framework, where copiotrophic *r*-strategists can grow faster on new carbon sources, thus outcompeting the oligotrophic *K*-strategists (121). Changes in the main benthic DOM producer are not exclusive to coral reefs and have been reported for coastal ecosystems worldwide, including macroalgae beds (130), kelp forests (131), and seagrass meadows (132). These macrophytes release significant amounts of their photosynthetically fixed carbon as DOM (38, 133, 134). Changes in benthic composition and resulting alterations of the local DOM pool may disrupt the stable microbial community states, inducing the rise of opportunistic heterotrophic microbes until a new equilibrium has been reached following the perturbation.

## Conclusion

Coral HMW DOM was compositionally distinct from ambient reef water and enriched opportunistic microbial taxa commonly associated with coral stress, significantly increasing the predicted metabolic potential for energy-, amino acid-, carbohydrate-, fatty acid and lipid-, and secondary metabolism (Fig. 7). In contrast, brown macroalgae HMW DOM was similar to ambient reef water and did not induce any effects on the bacterioplankton community composition. We propose two not mutually exclusive explanations for these results.

### *A greater alteration in HMW DOM composition through coral compared to algal exudates*

Our results indicate that whether coral or macroalgae DOM exerts stronger effects on the bacterioplankton community composition depends on local DOM and bacterioplankton characteristics which are at least partly shaped by the local (benthic) DOM-producing community. We hypothesize that a change in DOM away from the ambient composition acts as a disturbance, thus resulting in the dominance of opportunistic microbes that are able to adapt fast to environmental change.

### *A higher bacterial growth efficiency on coral compared to algal HMW exudates*

The strong effects of a small addition of coral HMW DOM to the bacterioplankton community suggest an efficient transformation of coral HMW DOM into microbial biomass, an important characteristic of nutrient cycles in healthy coral reefs. Brown macroalgae HMW exudate addition revealed no effects on the bacterioplankton community, indicating a low bacterial growth efficiency on algae exudates (i.e., more respiration). Brown algae HMW exudates, especially complex fucose-containing polysaccharides, could have additionally resisted microbial degradation (i.e., reduced bioavailability).

Overall, our results suggest that changes in HMW DOM composition support the rise of opportunistic microbes in coral reefs. Inefficient and/or incomplete degradation of HMW macroalgae exudates could ultimately lead to a reduced transfer of energy and nutrients stored in algal DOM to higher trophic levels, thereby supporting the proposed reef microbialization.

## ACKNOWLEDGMENTS

We thank Dr. Nicola Steinke and Dr. Silvia Vidal-Melgosa for advice on methodology and assistance with monosaccharide analysis and thank Tina Trautmann for conducting microarray analyses and sample preparations. We also thank Sven Pont for support with flow cytometry measurements and Karel Bakker for analyzing our inorganic nutrient samples. A special thanks goes to Pol Bosch and Reef Renewal Curaçao for providing *Acropora cervicornis* fragments from their coral nursery.

B.M.T. and C.W. received basis funding from the University of Bremen. B.M. received funding from the European Union's Horizon 2020 research and innovation program under the Marie Skłodowska-Curie grant (agreement No 894645). J.-H.H. received funding from the Cluster of Excellence initiative (EXC-2077-390741603) and the DFG Heisenberg program (HE 7217/1-1). This research was supported by the NWO award OCENW.M.21.178 to A.F.H.

## AUTHOR AFFILIATIONS

[1]Department of Marine Ecology, University of Bremen, Bremen, Germany

[2]Department of Marine Microbiology and Biogeochemistry, NIOZ Royal Netherlands Institute for Sea Research, Texel, Netherlands

[3]Department of Oceanography and Sea Grant College Program, Daniel K. Inouye Center for Microbial Oceanography: Research and Education, University of Hawaiʻi at Mānoa, Honolulu, Hawaiʻi, USA

[4]Marine Biology Research Division, Scripps Institute of Oceanography, University of California, San Diego, California, USA

[5]MARUM Center for Marine Environmental Sciences, University of Bremen, Bremen, Germany

[6]Department of Marine Glycobiology, Max Planck Institute for Marine Microbiology, Bremen, Germany

[7]Department of Freshwater and Marine Ecology, University of Amsterdam, Amsterdam, Netherlands

[8]CARMABI Foundation, Willemstad, Curaçao, Netherlands

## AUTHOR ORCIDs

Bianca M. Thobor  http://orcid.org/0000-0003-3862-1286
Andreas F. Haas  http://orcid.org/0000-0002-1150-8841
Christian Wild  http://orcid.org/0000-0001-9637-6536
Craig E. Nelson  http://orcid.org/0000-0003-2525-3496
Linda Wegley Kelly  http://orcid.org/0000-0002-9157-4426
Jan-Hendrik Hehemann  http://orcid.org/0000-0002-8700-2564
Milou G. I. Arts  http://orcid.org/0000-0002-7245-5034
Hagen Buck-Wiese  http://orcid.org/0000-0002-4807-5795
Benjamin Mueller  http://orcid.org/0000-0001-9335-0437

## FUNDING

| Funder | Grant(s) | Author(s) |
|---|---|---|
| EC \| H2020 \| PRIORITY 'Excellent science' \| H2020 Marie Skłodowska-Curie Actions (MSCA) | 894645 | Benjamin Mueller |

| Funder | Grant(s) | Author(s) |
|---|---|---|
| Deutsche Forschungsgemeinschaft (DFG) | EXC-2077-390741603 | Jan-Hendrik Hehemann |
| Deutsche Forschungsgemeinschaft (DFG) | HE 7217/1-1 | Jan-Hendrik Hehemann |
| Nederlandse Organisatie voor Wetenschappelijk Onderzoek (NWO) | OCENW.M.21.178 | Andreas F. Haas |
| Universität Bremen (Uni Bremen) | | Bianca Maria Thobor |
| Universität Bremen (Uni Bremen) | | Christian Wild |

## AUTHOR CONTRIBUTIONS

Bianca M. Thobor, Conceptualization, Data curation, Formal analysis, Investigation, Methodology, Visualization, Writing – original draft | Andreas F. Haas, Conceptualization, Funding acquisition, Resources, Supervision, Writing – review and editing | Christian Wild, Conceptualization, Funding acquisition, Resources, Supervision, Writing – review and editing | Craig E. Nelson, Data curation, Methodology, Resources, Software, Writing – review and editing | Linda Wegley Kelly, Methodology, Writing – review and editing | Jan-Hendrik Hehemann, Funding acquisition, Resources, Supervision, Writing – review and editing | Milou G. I. Arts, Investigation, Writing – review and editing | Meine Boer, Data curation, Software, Writing – review and editing | Hagen Buck-Wiese, Methodology, Writing – review and editing | Nguyen P. Nguyen, Methodology, Writing – review and editing | Inga Hellige, Investigation, Methodology, Writing – review and editing | Benjamin Mueller, Conceptualization, Investigation, Methodology, Supervision, Writing – review and editing

## DATA AVAILABILITY

Raw sequence reads determined in this study are available at NCBI under the BioProject accession number PRJNA1163255. Raw data of genus-level microbial community compositions (Fig. 5), predicted metabolic functions (Fig. 6), and R codes for the analysis and visualization of the respective data sets are available online in the zenodo data repository (https://doi.org/10.5281/zenodo.13365525). Remaining raw data (Fig. 2–5) are available in the supplemental material.

## ETHICS APPROVAL

All collections and experimental work were carried out under the research permit (#2012/48584) issued by the Curaçaoan Ministry of Health, Environment and Nature (GMN) to the CARMABI foundation. Research only included animals of lower taxonomic ranks (i.e., Cnidaria) and macroalgae, which do not require approval by an ethics committee according to German § 5 TierSchG (4 July 2013) and the European Directive 2010/63/EU (22 September 2010).

## ADDITIONAL FILES

The following material is available online.

### Supplemental Material

**Supplemental Material (mSystems00832-24-S0001.pdf).** Supplemental methods, Fig. S1–S7, and Tables S1–S8.

### Open Peer Review

**PEER REVIEW HISTORY (review-history.pdf).** An accounting of the reviewer comments and feedback.

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
