## [Reviewer comments · mSystems]

Coral high molecular weight carbohydrates support opportunistic microbes in bacterioplankton from an algae-dominated reef

Bianca Thobor, Andreas Haas, Christian Wild, Craig Nelson, Linda Wegley Kelly, Jan-Hendrik Hehemann, Milou Arts, Meine Boer, Hagen Buck-Wiese, Nguyen Nguyen, Inga Hellige, and Benjamin Mueller

Corresponding Author(s): Bianca Thobor, Universitat Bremen

Review Timeline:

Submission Date:	June 19, 2024
Editorial Decision:	August 15, 2024
Revision Received:	August 26, 2024
Editorial Decision:	September 10, 2024
Revision Received:	September 24, 2024
Accepted:	September 24, 2024

Editor: Julie Meyer

Reviewer(s): The reviewers have opted to remain anonymous.

Transaction Report:

DOI: <https://doi.org/10.1128/msystems.00832-24>

Re: mSystems00832-24 (Coral high molecular weight carbohydrates support opportunistic microbes in bacterioplankton from an algae-dominated reef)

Dear Dr. Bianca Maria Thobor:

In accordance with ASM's data policy, you will need to add a Data Availability paragraph at the end of the Methods section. The Data Availability paragraph should include a data description, name(s) of the repositories, and digital object identifiers (DOIs) or accession numbers. Additional details about the data sharing policy are found here: <https://journals.asm.org/open-data-policy>.

Revision Guidelines

Sincerely,
Julie Meyer
Editor
mSystems

Reviewer #1 (Comments for the Author):

In this work, Thobor et al. use aquaria-based studies to investigate the effect of high molecular weight fraction exudates from corals and algae sourced from an alga dominated reef on bacterioplankton growth. In contrast to previous studies, they observe

growth of pathogenic/harmful microbes in coral exudates as opposed to algal exudates. To explain these observations, a series of well-designed analytics is performed. The reviewer has very minor suggestions for discussion.

Overall, the manuscript is extremely well and clearly written. All the statistics used are clearly explained, appropriate, and described in detail in the methods section. The discussion is elaborate and appropriately discusses, contrasts and provides multiple explanations for the contrasting results observed. The only limitation of manuscript is that the functional prediction is done via 16s analysis and not metagenomics. This limitation is mentioned by the authors but should also be stated again in discussion. Functional prediction aligns with the rest of the findings in the manuscript. Thus, the reviewer is not too worried about the scope of the manuscript being affected by this limitation. Lastly, it is both HMW and LMW fractions that would affect the growth of bacterioplankton. The results are contrasting from previously published literature because of the difference in fraction used and the effect of dilution. Thus, future insights/proposed work should be provided where these fractions are mixed in naturally occurring proportions from both algal dominated and coral dominated reefs to study their combined effect on bacterioplankton growth and composition.

Reviewer #2 (Comments for the Author):

The manuscript by Thobar et al describes a well-designed experiment focused on the impact of HMW DOM from corals, algae, or background seawater on the bacterioplankton community. This is the first to isolate HMW and use that as an addition to plankton incubation experiments. The experimental design and data analysis are sound and the results have relevant implications for coral reef ecosystem ecology and microbial ecology more broadly. I have minor comments and questions for the authors that I will list below. The only big picture comment (and it is also listed below) is that I think there is room in the interpretation for DOM of varying composition in general to have multifaceted impacts on the picoplankton community and interpreting these impacts is not necessarily straightforward. I think it is still possible to make cautious generalizations as the authors currently have here.

- L82: To be clearer with this sentence, the Nelson et al. study profiled carbohydrates as part of LMW or HMW?
- L155-162: This part confused me until I realized the seawater collection described here is for the control incubation - is that correct? It would be helpful to state that.
- L161: Does "The tank.." refer to the incubation tanks - so 3 for each treatment and 4 for the controls?
- L200: Was the seawater used in the incubations as the base ("medium") filtered to remove large grazers? If not, why?
- L202: It does give some volumes later on but for ease of reading it would be great to give the volume that was sampled here for each of the five timepoints
- L370: It would be helpful to add in coral result from fig2c
- L443: I expected the taxa name since it is only one group - maybe add in OCS116 group
- L473-476: This is for coral still yes?
- L476-479: It would be helpful to talk about the control from Fig 6 carbohydrates here as well
- L553-554: This sentence makes sense, but I suggest adding in some way that the difference of concentration vs composition of LMW not able to uncouple at this point in time.
- L585: Should 'alternative stable state' instead be 'phase shift' to be consistent with the literature which still has some discussion on this topic (e.g., Dudgeon et al. 2010 MEPS; van de Leemput et al. 2016 Coral Reefs; Crisp et al. 2022 Mar Env Res)?
- L590: This sounds completely reasonable, but it is also possible that this phenomenon/impact is more nuanced than previous studies have suggested and the results are more complicated to interpret than we previously thought.
- L592-593: for the respiration, sounds reasonable and would need to test in this specific system yes?
- L594: Fucoidan was measured at start and end as in the supplemental yes? It sounds like this could be clearer if using support from this experiment or not.

Figures: Figures are clear and well labeled. One thing I suggest is to add that the end point is 4 days in the figure captions.

Ref. number: mSystems00832-24

Manuscript title: Coral high molecular weight carbohydrates support opportunistic microbes in bacterioplankton from an algae-dominated reef

We thank the reviewers for their constructive feedback on our manuscript. Our replies are shown in blue below each comment, and quotation marks indicate when text was directly copied from the improved manuscript.

Editor comments

In accordance with ASM's data policy, you will need to add a Data Availability paragraph at the end of the Methods section. The Data Availability paragraph should include a data description, name(s) of the repositories, and digital object identifiers (DOIs) or accession numbers. Additional details about the data sharing policy are found here: <https://journals.asm.org/open-data-policy>.

We have added a data availability statement at the end of the methods section as requested (lines 362-366):

“Data availability

Raw data of genus-level microbial community compositions (Fig. 5), predicted metabolic functions (Fig. 6), and R codes for the analysis and visualization of the respective datasets are available online in the zenodo data repository (<https://doi.org/10.5281/zenodo.13365525>). Remaining raw data (Fig. 2-5) is available in the supplementary material.”

We also added in lines 143-145 the required statement on the use of animals (under “detailed information” in the submission system):

“Research only included animals of lower taxonomic ranks (i.e. Cnidaria) and macroalgae, which do not require approval by an ethics committee according to German § 5 TierSchG (04.07.2013) and the European Directive 2010/63/EU (22.09.2010).”

We added all figure legends to the end of the manuscript text file and uploaded all images as .eps files as requested.

Reviewer #1 (Comments for the Author):

In this work, Thobor et al. use aquaria-based studies to investigate the effect of high molecular weight fraction exudates from corals and algae sourced from an alga dominated reef on bacterioplankton growth. In contrast to previous studies, they observe growth of pathogenic/harmful microbes in coral exudates as opposed to algal exudates. To explain these observations, a series of well-designed analytics is performed. The reviewer has very minor suggestions for discussion.

Overall, the manuscript is extremely well and clearly written. All the statistics used are clearly explained, appropriate, and described in detail in the methods section. The discussion is elaborate and appropriately discusses, contrasts and provides multiple explanations for the contrasting results observed. The only limitation of manuscript is that the functional prediction is done via 16s analysis and not metagenomics. **This limitation is mentioned by the authors but should also be stated again in discussion.** Functional prediction aligns with the rest of the findings in the manuscript. Thus, the reviewer is not too worried about the scope of the manuscript being affected by this limitation.

We agree and added to lines 501-504:

“Furthermore, **although the prediction of metabolic functions from 16S rRNA data is limited in accuracy** (1, 2) (see **Methods section for details**), the increase in predicted carbohydrate-, amino acid-, and fatty acid and lipid metabolism in bacterioplankton communities (Fig. 6) is consistent with degradation of coral mucus, which mainly consists of carbohydrates, proteins, and lipids (3, 4). “

Lastly, it is both HMW and LMW fractions that would affect the growth of bacterioplankton. The results are **contrasting from previously published literature because of the difference in fraction used and the effect of dilution**. Thus, **future insights/proposed** work should be provided where these fractions are mixed in naturally occurring proportions from both algal dominated and coral dominated reefs to study their combined effect on bacterioplankton growth and composition.

We agree and added to lines 550-552:

“**Further studies using undiluted bacterioplankton communities in combination with a natural ratio of HMW and LMW coral and macroalgal DOM are required to fully unravel the control of DOM source (and thereby composition) on bacterioplankton communities.**”

Reviewer #2 (Comments for the Author):

The manuscript by Thobor et al describes a well-designed experiment focused on the impact of HMW DOM from corals, algae, or background seawater on the bacterioplankton community. This is the first to isolate HMW and use that as an addition to plankton incubation experiments. The experimental design and data analysis are sound and the results have relevant implications for coral reef ecosystem ecology and microbial ecology more broadly. I have minor comments and questions for the authors that I will list below. The only big picture comment (and it is also listed below) is that I think there is room in the interpretation for DOM of varying composition in general to have multifaceted impacts on the picoplankton community and interpreting these impacts is not necessarily straightforward. I think it is still possible to make cautious generalizations as the authors currently have here.

L82: To be clearer with this sentence, the Nelson et al. study profiled carbohydrates as part of LMW or HMW?

The study by Nelson et al. did not use DOM size fractionation and instead added bulk DOM to the microbial communities (thereby diluting the initial microbial community as well as increasing the DOM concentration considerably, with 1.4 - 2.3 times higher enrichment for algal compared to coral DOM. This is also discussed in the next paragraph). We clarified this as follows (lines 80-85):

“Nelson et al. (5) found increased concentrations of fucose, galactose, and rhamnose in **bulk** macroalgae-DOM (i.e., **HMW + LMW**), which exerted strong effects on the composition of natural bacterioplankton communities (5). Particularly the growth of copiotrophs and putative pathogens belonging to *Gammaproteobacteria* was stimulated. In contrast, **bulk** DOM exuded by the coral *Porites lobata* mainly consisted of glucose, mannose, and xylose and was similar in composition to the ambient reef water.”

L155-162: This part confused me until I realized the seawater collection described here is for the control incubation - is that correct? It would be helpful to state that.

Here, we describe the collection and filtration of seawater for all experimental incubations. The initial water in aquaria was always collected and filtered the same way and the only difference was that we placed corals, algae, or no benthic primary producer (controls) in the aquaria with filtered seawater. We clarified this (lines 155-159):

“One day prior to the experiment (i.e., **control-, algae- or coral incubations**), seawater was directly collected from the inlet pipe and run over filters with decreasing pore sizes to remove particles (50, 20, 5, 0.5 μm) and microbes (0.2 μm , 12 cm diameter, polycarbonate, pre-flushed with 1 L seawater), for 4 hours.”

L161: Does "The tank.." refer to the incubation tanks - so 3 for each treatment and 4 for the controls?

Correct. It is phrased in singular because we used one experimental setup for all incubations which were done on separate days (as described at the start of the paragraph). We clarified here (lines 161-162):

“The **incubation** tank was placed in a flow-through water bath with fresh seawater to keep temperatures close to *in situ* conditions (measured every five minutes with HOBO Pendant; Table 1).”

L200: Was the seawater used in the incubations as the base ("medium") filtered to remove large grazers? If not, why?

We did not remove grazers due to several reasons. To clarify, we added in line 194-199:

“**We decided to use unfiltered seawater (i.e., without removing potential grazers) to include larger aggregates and/or colloidal material, which are important hotspots of planktonic microbial diversity on coral reefs (6–8) to assess the effects of HMW DOM on bacterioplankton communities under natural conditions. Furthermore, prefiltration reduces grazing without concomitantly reducing viral lysis which may influence microbial community compositions (9).**”

L202: It does give some volumes later on but for ease of reading it would be great to give the volume that was sampled here for each of the five timepoints

We agree and added in lines 206-208:

“Samples for all parameters were collected at five timepoints: shortly after the start of the incubations (ca. 30 minutes after exudate addition, **1.3 L**), after six and 12 hours (**0.3 L each**), and after one and four days (**1.3 L each**).”

L370: It would be helpful to add in coral result from fig2c

We agree and added in lines 374-377:

“Combined HMW carbohydrate concentrations were significantly enriched by 168 % in macroalgae incubations compared to seawater controls and starting concentrations (ANOVA, $F_{(3,9)} = 6.5$, $p < 0.05$, $\eta^2_G = 0.68$, HSD test; Fig. 2c), **while coral incubations were not significantly enriched (Fig. 2c).**”

L443: I expected the taxa name since it is only one group - maybe add in OCS116 group

To not repeat information from the figure, we decided to not list the genus names which were enriched (black names in Fig. 5c), but rather state which class and family they belonged to. The OCS116 group was not enriched with coral exudates, but depleted with algae exudates with random forest analysis (see green bar going to the left in the left panel of Fig. 5c).

L473-476: This is for coral still yes?

Correct. The addition of coral HMW DOM did not affect the cell density or nutrient concentrations, only the bacterioplankton community composition. See lines 487-490:

“The compositional differences in exuded HMW DOM, added at low concentrations (< 1 % of ambient DOC), had no effect on the overall bacterioplankton cell density or dissolved nutrient concentrations (Supplementary Fig. S3).”

L476-479: It would be helpful to talk about the control from Fig 6 carbohydrates here as well

We compared the predicted metabolic functions to controls for both treatments. To clarify we added to line 490-493:

“However, coral HMW DOM significantly affected the bacterioplankton community composition (Fig. 5) and increased the predicted potential for specific metabolic functions **relative to controls** (Fig. 6). In contrast, algae HMW DOM addition induced no significant differences compared to seawater controls.”

L553-554: This sentence makes sense, but I suggest adding in some way that the difference of concentration vs composition of LMW not able to uncouple at this point in time.

We agree and added to lines 559-562:

“Our approach thereby allowed a decoupling of DOM concentration-dependent from DOM composition-dependent effects, which suggests that the increased DOC concentration component at least partly (**excluding potential effects from LMW DOM**) explained previously reported differences in bacterial growth rates between coral and algae DOM.”

L585: Should 'alternative stable state' instead be 'phase shift' to be consistent with the literature which still has some discussion on this topic (e.g., Dudgeon et al. 2010 MEPS; van de Leemput et al. 2016 Coral Reefs; Crisp et al. 2022 Mar Env Res)?

We prefer the use of “alternative stable state” here, as we are not referring to an ongoing shift (i.e., phase shift), but rather to an ecosystem that already “shifted” and which is currently in alternative stable state (i.e., macroalgae-dominated reef). This term is also commonly used (10, 11).

L590: This sounds completely reasonable, but it is also possible that this phenomenon/impact is more nuanced than previous studies have suggested and the results are more complicated to interpret than we previously thought.

Thank you for this comment. We agree that the interpretation of results and reality in general is usually highly complex. Nevertheless, we are convinced that there is value in a certain level of simplification to gain a better understanding of underlying processes. We therefore formulated a new hypothesis based on our results yet took caution to clearly identify it as such by introducing the statement with “we therefore propose” and the use of “may”.

L592-593: for the respiration, sounds reasonable and would need to test in this specific system yes?

Correct. We agree and added to clarify (lines 598-600):

“Brown macroalgae HMW DOM did not appear to support microbial growth, which could be explained by increased respiration instead of biomass production (**not measured here but shown previously** (12–14))”

L594: Fucoidan was measured at start and end as in the supplemental yes? It sounds like this could be clearer if using support from this experiment or not.

We did not measure fucoidan in the HMW macroalgal DOM, but in the tissue of the used macroalgae. We tried it on HMW algae DOM concentrate as well, however with a concentration factor of 100 times it was not successful. We assume that a much higher concentration factor of at least ~ 2500 (see (15) 100 L -> 40 mL) would be necessary yet wasn't feasible in our study. However, the monosaccharide (i.e., building blocks of glycans) composition of extracted fucoidan was similar to the

monosaccharide composition of algal HMW DOM (see lines 570-573) and ambient reef water (lines 577-580). Thus, we hypothesize, that based on the similarity in monosaccharide composition, fucoidan was part of the macroalgal and seawater HMW DOM fraction and was therefore added to bacterioplankton communities in the experiment. We consider this a valid assumption and subsequently discuss the ecological implications this may have.

Figures: Figures are clear and well labeled. One thing I suggest is to add that the end point is 4 days in the figure captions.

We agree and added "... after four days (**final timepoint**)" to the legends of Fig. 5 (line 1046) & 6 (line 1066).

References

1. Douglas GM, Maffei VJ, Zaneveld JR, Yurgel SN, Brown JR, Taylor CM, Huttenhower C, Langille MGI. 2020. PICRUSt2 for prediction of metagenome functions. *Nat Biotechnol* 38:685–688.
2. Mongad DS, Chavan NS, Narwade NP, Dixit K, Shouche YS, Dhotre DP. 2021. MicFunPred: A conserved approach to predict functional profiles from 16S rRNA gene sequence data. *Genomics* 113:3635–3643.
3. Crossland CJ. 1987. *In situ* release of mucus and DOC-lipid from the corals *Acropora variabilis* and *Stylophora pistillata* in different light regimes. *Coral Reefs* 6:35–42.
4. Ducklow HW, Mitchell R. 1979. Composition of mucus released by coral reef coelenterates. *Limnology and Oceanography* 24:706–714.
5. Nelson CE, Goldberg SJ, Wegley Kelly L, Haas AF, Smith JE, Rohwer F, Carlson CA. 2013. Coral and macroalgal exudates vary in neutral sugar composition and differentially enrich reef bacterioplankton lineages. *ISME J* 7:962–979.
6. Huettel M, Wild C, Gonelli S. 2006. Mucus trap in coral reefs: formation and temporal evolution of particle aggregates caused by coral mucus. *Mar Ecol Prog Ser* 307:69–84.
7. Naumann M, Richter C, el-Zibdah M, Wild C. 2009. Coral mucus as an efficient trap for picoplanktonic cyanobacteria: implications for pelagic–benthic coupling in the reef ecosystem. *Mar Ecol Prog Ser* 385:65–76.
8. McNally SP, Parsons RJ, Santoro AE, Apprill A. 2017. Multifaceted impacts of the stony coral *Porites astreoides* on picoplankton abundance and community composition. *Limnology and Oceanography* 62:217–234.
9. Cram JA, Parada AE, Fuhrman JA. 2016. Dilution reveals how viral lysis and grazing shape microbial communities. *Limnology and Oceanography* 61:889–905.
10. Nyström M, Graham NAJ, Lokrantz J, Norström AV. 2008. Capturing the cornerstones of coral reef resilience: linking theory to practice. *Coral Reefs* 27:795–809.
11. van de Leemput IA, Hughes TP, van Nes EH, Scheffer M. 2016. Multiple feedbacks and the prevalence of alternate stable states on coral reefs. *Coral Reefs* 35:857–865.
12. Haas AF, Fairouz MFM, Kelly LW, Nelson CE, Dinsdale EA, Edwards RA, Giles S, Hatay M, Hisakawa N, Knowles B, Lim YW, Maughan H, Pantos O, Roach TNF, Sanchez SE, Silveira

- CB, Sandin S, Smith JE, Rohwer F. 2016. Global microbialization of coral reefs. *Nat Microbiol* 1:16042.
13. Haas AF, Nelson CE, Rohwer F, Wegley-Kelly L, Quistad SD, Carlson CA, Leichter JJ, Hatay M, Smith JE. 2013. Influence of coral and algal exudates on microbially mediated reef metabolism. *PeerJ* 1:e108.
 14. Mueller B, Brocke HJ, Rohwer FL, Dittmar T, Huisman J, Vermeij MJA, de Goeij JM. 2022. Nocturnal dissolved organic matter release by turf algae and its role in the microbialization of reefs. *Functional Ecology* 36:2104–2118.
 15. Vidal-Melgosa S, Sichert A, Francis TB, Bartosik D, Niggemann J, Wichels A, Willats WGT, Fuchs BM, Teeling H, Becher D, Schweder T, Amann R, Hehemann J-H. 2021. Diatom fucan polysaccharide precipitates carbon during algal blooms. 1. *Nature Communications* 12:1150.

Re: mSystems00832-24R1 (Coral high molecular weight carbohydrates support opportunistic microbes in bacterioplankton from an algae-dominated reef)

Dear Dr. Bianca Maria Thobor:

I am requesting minor modifications so that you can update the information on data deposition. Per the email of last week, I have reviewed your revised manuscript, and I am ready to accept it without additional peer reviews. Thank you for incorporating the minor recommendations of the reviewers and depositing data and code in Zenodo. Since this manuscript includes the generation of nucleotide sequence data, the raw sequencing data will need to be deposited in a repository like NCBI's Sequence Read Archive or EMBL Nucleotide Sequence Data Base to comply with ASM's data policy: <https://journals.asm.org/open-data-policy>

As soon as you have incorporated an accession number for the sequencing data, I will be happy to accept the final version of the manuscript. I look forward to seeing your manuscript in an upcoming issue of mSystems.

Revision Guidelines

Sincerely,
Julie Meyer
Editor
mSystems

Ref. number: mSystems00832-24

Manuscript title: Coral high molecular weight carbohydrates support opportunistic microbes in bacterioplankton from an algae-dominated reef

Editor comments

I am requesting minor modifications so that you can update the information on data deposition. Per the email of last week, I have reviewed your revised manuscript, and I am ready to accept it without additional peer reviews. Thank you for incorporating the minor recommendations of the reviewers and depositing data and code in Zenodo. Since this manuscript includes the generation of nucleotide sequence data, the raw sequencing data will need to be deposited in a repository like NCBI's Sequence Read Archive or EMBL Nucleotide Sequence Data Base to comply with ASM's data policy: <https://journals.asm.org/open-data-policy>

As soon as you have incorporated an accession number for the sequencing data, I will be happy to accept the final version of the manuscript. I look forward to seeing your manuscript in an upcoming issue of mSystems.

Dear Dr. Julie Meyer,

Thank you for the fast decision on our manuscript. We have now uploaded the raw sequence data to NCBI under the BioProject number PRJNA1163255, indicated in lines 365-366 of the revised manuscript:

“Raw sequence reads determined in this study are available at NCBI under the BioProject accession number PRJNA1163255.”

Sincerely,

Bianca Thobor

Re: mSystems00832-24R2 (Coral high molecular weight carbohydrates support opportunistic microbes in bacterioplankton from an algae-dominated reef)

Dear Dr. Bianca Maria Thobor:

Your manuscript has been accepted, and I am forwarding it to the ASM production staff for publication. Your paper will first be checked to make sure all elements meet the technical requirements. ASM staff will contact you if anything needs to be revised before copyediting and production can begin. Otherwise, you will be notified when your proofs are ready to be viewed.

Sincerely,
Julie Meyer
Editor
mSystems